# Changes in global and thalamic brain connectivity in LSD-induced altered states of consciousness are attributable to the 5-HT2A receptor

Katrin H Preller[1,2]*, Joshua B Burt[2,3], Jie Lisa Ji[2], Charles H Schleifer[2], Brendan D Adkinson[2], Philipp Stämpfli[4], Erich Seifritz[4], Grega Repovs[5], John H Krystal[2], John D Murray[2,3,6], Franz X Vollenweider[1†], Alan Anticevic[2†]

[1]Neuropsychopharmacology and Brain Imaging, Department of Psychiatry, Psychotherapy and Psychosomatics, University Hospital for Psychiatry Zurich, Zurich, Switzerland; [2]Department of Psychiatry, Yale University School of Medicine, New Haven, United States; [3]Department of Physics, Yale University, New Haven, United States; [4]Department of Psychiatry, Psychotherapy and Psychosomatics, University Hospital for Psychiatry Zurich, Zurich, Switzerland; [5]Mind and Brain Lab, Department of Psychology, University of Ljubljana, Ljubljana, Slovenia; [6]Department of Neuroscience, Yale University School of Medicine, New Haven, United States

*For correspondence:
preller@bli.uzh.ch

†These authors contributed equally to this work

## Abstract

**Background:** Lysergic acid diethylamide (LSD) has agonist activity at various serotonin (5-HT) and dopamine receptors. Despite the therapeutic and scientific interest in LSD, specific receptor contributions to its neurobiological effects remain unknown.
**Methods:** We therefore conducted a double-blind, randomized, counterbalanced, cross-over studyduring which 24 healthy human participants received either (i) placebo+placebo, (ii) placebo+LSD (100 µg po), or (iii) Ketanserin, a selective 5-HT2A receptor antagonist,+LSD. We quantified resting-state functional connectivity via a data-driven global brain connectivity method and compared it to cortical gene expression maps.
**Results:** LSD reduced associative, but concurrently increased sensory-somatomotor brain-wide and thalamic connectivity. Ketanserin fully blocked the subjective and neural LSD effects. Whole-brain spatial patterns of LSD effects matched 5-HT2A receptor cortical gene expression in humans.
**Conclusions:** Together, these results strongly implicate the 5-HT2A receptor in LSD's neuropharmacology. This study therefore pinpoints the critical role of 5-HT2A in LSD's mechanism, which informs its neurobiology and guides rational development of psychedelic-based therapeutics.
**Funding:** Funded by the Swiss National Science Foundation, the Swiss Neuromatrix Foundation, the Usona Institute, the NIH, the NIAA, the NARSAD Independent Investigator Grant, the Yale CTSA grant, and the Slovenian Research Agency.
**Clinical trial number:** NCT02451072.

## Introduction

Disorders of perception and the form and content of thought are important contributors to the global burden of disease (*Murray et al., 2012*). Mechanistic studies of consciousness may be undertaken using psychedelic drugs as pharmacologic probes of molecular signaling within cortical networks underlying perception and thought. In particular, lysergic acid diethylamide (LSD) is a psychedelic drug with predominantly agonist activity at serotonin (5-HT)2A/C, −1A/B, −6, and −7

**eLife digest** The psychedelic drug LSD alters thinking and perception. Users can experience hallucinations, in which they, for example, see things that are not there. Colors, sounds and objects can appear distorted, and time can seem to speed up or slow down. These changes bear some resemblance to the changes in thinking and perception that occur in certain psychiatric disorders, such as schizophrenia. Studying how LSD affects the brain could thus offer insights into the mechanisms underlying these conditions. There is also evidence that LSD itself could help to reduce the symptoms of depression and anxiety disorders.

Preller et al. have now used brain imaging to explore the effects of LSD on the brains of healthy volunteers. This revealed that LSD reduced communication among brain areas involved in planning and decision-making, but it increased communication between areas involved in sensation and movement. Volunteers whose brains showed the most communication between sensory and movement areas also reported the strongest effects of LSD on their thinking and perception.

Preller et al. also found that another drug called Ketanserin prevented LSD from altering how different brain regions communicate. It also prevented LSD from inducing changes in thinking and perception. Ketanserin blocks a protein called the serotonin 2A receptor, which is activated by a brain chemical called serotonin that, amongst other roles, helps to regulate mood. By mapping the location of the gene that produces the serotonin 2A receptor, Preller et al. showed that the receptor is present in brain regions that show altered communication after LSD intake, therefore pinpointing the importance of this receptor in the effects of LSD.

Psychiatric disorders that produce psychotic symptoms affect vast numbers of people worldwide. Further research into how LSD affects the brain could help us to better understand how such symptoms arise, and may also lead to the development of more effective treatments for a range of mental health conditions.

and dopamine D2 and D1 receptors (R). Its administration produces characteristic alterations in perception, mood, thought, and the sense of self (*Marona-Lewicka et al., 2002*; *Nichols, 2004*). Despite its powerful effects on consciousness, human research on LSD neurobiology stalled in the late 1960s because of a narrow focus on the experiential effects of hallucinogenic drugs, combined with a lack of understanding of its effects on molecular signaling mechanisms in the brain. However, renewed interest in the potentially beneficial clinical effects of psychedelics (*Carhart-Harris et al., 2016a*; *Gasser et al., 2014*; *Griffiths et al., 2016*) warrants a better understanding of their underlying neuropharmacology. Nevertheless, major knowledge gaps remain regarding LSD's neurobiology in humans as well as its time-dependent receptor neuropharmacology.

To address this critical gap, the current study aims to comprehensively map time-dependent pharmacological effects of LSD on neural functional connectivity in healthy human adults and compare them to the spatial expression profile of genes coding for receptors interacting with LSD. The goal is to leverage the statistical properties of the slow (<1 Hz) intrinsic fluctuations of the blood-oxygen-level-dependent (BOLD) signal hemodynamics at rest (i.e. resting-state functional connectivity (rs-fcMRI)). Critically, rs-fcMRI analyses are able to reveal the functional architecture of the brain, which is organized into large-scale systems exhibiting functional relationships across space and time (*Biswal et al., 2010*; *Buckner et al., 2013*; *Yang et al., 2014*). Rs-fcMRI measures have furthermore revealed potential biomarkers of various neural disorders (*Murrough et al., 2016*; *Yang et al., 2016a*), as well as proven sensitive to the effects of neuropharmacological agents (*Driesen et al., 2013a*; *Anticevic et al., 2015*).

Focused analyses on specific regions revealed effects of intravenously administered LSD on functional connectivity between V1 and distributed cortical and subcortical regions (*Carhart-Harris et al., 2016b*). However, such 'seed-based' approaches rely on explicitly selecting specific regions of interest based on a priori hypotheses. Therefore, such an approach has limited ability to detect pharmacologically-induced dysconnectivity not predicted a priori. To characterize LSD effects on functional connectivity in the absence of strong a priori hypotheses, the current study employed a fully data-driven approach derived from graph theory called Global Brain Connectivity (GBC) (*Anticevic et al., 2014b*). In essence, GBC computes the connectivity of every voxel in the

brain with all other voxels and summarizes that in a single value. Therefore, areas of high GBC are highly functionally connected with other areas and might play a role in coordinating large-scale patterns of brain activity (*Cole et al., 2010*). Reductions in GBC may indicate decreased participation of a brain area in larger networks, whereas increased GBC may indicate a broadening or synchronization of functional networks (*Anticevic et al., 2014b*). One focused study examined GBC after intravenously administered LSD in a sample of 15 participants, revealing connectivity elevations across higher-order association cortices (*Tagliazucchi et al., 2016*). While compelling, this preliminary study did not take into account the influence of global signal (GS) artefacts (e.g. via global signal regression, GSR), which are known to exhibit massive differences in clinical populations and following pharmacological manipulations (*Power et al., 2017*; *Yang et al., 2016b*; *Lewis et al., 2017*; *Driesen et al., 2013b*). Specifically, GS is hypothesized to contain a complex mixture of non-neuronal artefacts (e.g., physiological, movement, scanner-related), which can induce elevated relationships across the brain (*Yang et al., 2014*). No study has examined LSD-induced changes as a function of GS removal. To inform this knowledge gap a major objective here was to study data-driven LSD-induced dysconnectivity in the context of GS removal.

Another aim of the current study was to determine the extent to which the neural and behavioral effects of LSD are mediated by 5-HT$_{2A}$ receptors. Preclinical studies suggest that LSD binds potently to many neuroreceptors including 5-HT$_{2A}$, 5-HT$_{2C}$, 5-HT$_{1A}$, D2, and other receptors (*Marona-Lewicka et al., 2002*; *Passie et al., 2008*). Yet, a recent paper from our group (*Preller et al., 2017*) reported that the psychedelic effects of LSD were entirely blocked in humans by ketanserin (Ket), a selective antagonist at 5-HT$_{2A}$ and α-adreno receptors (*Leysen et al., 1982*). This would suggest that the neural effects of LSD should be blocked by Ket. It also suggests that networks modulated by LSD should highly be associated with the distribution of 5-HT$_{2A}$ receptors in the brain and not closely associated with the distribution of receptors unrelated to the mechanism of action of LSD.

Here we leverage recent advances (*Burt et al., 2018*) in human cortical gene expression mapping to inform the spatial topography of neuropharmacologically-induced changes in data-driven connectivity. We hypothesized that the LSD-induced GBC change will quantitatively match the spatial expression profile of genes coding for the 5-HT$_{2A}$ receptor. In turn, we hypothesized that this effect will be preferential for the 5-HT$_{2A}$ but not other receptors and that the spatial match will be vastly improved after artefact removal. In doing so, this convergence of neuropharmacology and gene expression mapping validates the contribution of the 5-HT$_{2A}$ receptor to LSD neuropharmacology. In turn, it also highlights a general method for relating spatial gene expression profiles to neuropharmacological manipulations, which has direct and important implications for the rational refinement of any receptor neuropharmacology.

Collectively, this pharmacological neuroimaging study addresses the following major knowledge gaps in our understanding of LSD neurobiology, by demonstrating: (i) data-driven LSD effects across brain-wide networks, which are exquisitely sensitive to GS removal, (ii) the subjective and neural effects of LSD neuropharmacology are attributable to the 5-HT$_{2A}$ receptor, and (iii) the cortex-wide LSD effects can be mapped onto the spatial expression profile of the gene coding for the 5-HT$_{2A}$ receptor.

## Results

### LSD modulates global brain connectivity and induces marked subjective drug effects

The main effect of drug on GBC computed with global signal regression (GSR) revealed significant (TFCE type I error protected, 10000 permutations) widespread differences in GBC between drug conditions in cortical and subcortical areas (*Figure 1*). Comparing LSD to Ketanserin+LSD (Ket+LSD) +Placebo (Pla) conditions across sessions shows that LSD induces hyper-connectivity predominately in sensory and somatomotor areas, that is the occipital cortex, the superior temporal gyrus, and the postcentral gyrus, as well as the precuneus. Hypo-connectivity was induced in subcortical areas as well as cortical areas associated with associative networks, such the medial and lateral prefrontal cortex, the cingulum, the insula, and the temporoparietal junction. All changes in connectivity were expressed bilaterally (*Figure 1A*). *Figure 1B* shows mean connectivity strength (Fz) for each drug condition and the distribution of Fz values for grayordinates (i.e. either a surface vertex (node) or a

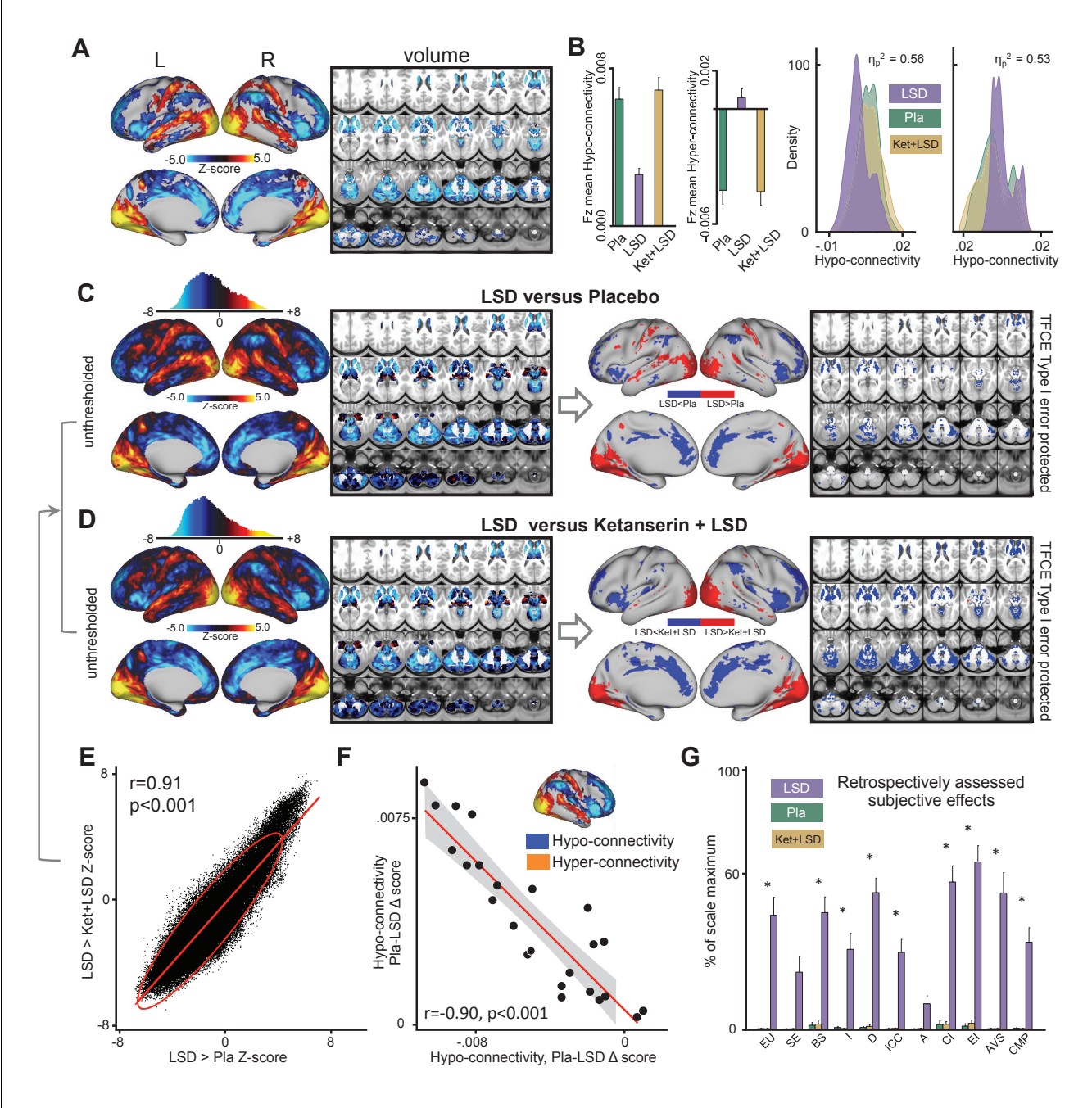

**Figure 1.** Effect of drug condition on global brain connectivity and subjective drug effects. (**A**) Z-score map for the effect of LSD condition vs. (Ket +LSD)+Pla condition within areas showing a significant main effect of drug (TFCE type I error protected). Red/orange areas indicate regions where participants exhibited stronger GBC in the LDS condition, whereas blue areas indicate regions where participants exhibited reduced GBC condition, compared with (Ket+LSD)+Pla conditions. (**B**) Bar plots show mean connectivity strength (Fz) values for hyper- and hypo-connected areas averaged across grayordinates showing a significant main effect of drug. Distribution plots show distribution of Fz values within grayordinates showing significant hyper- and hypo-connectivity for LSD compared to (Ket+LSD)+Pla conditions. (**C**) Right panel displays significant (TFCE type I error protected) areas showing increased (red) and decreased (blue) GBC in the LSD condition compared to Pla. Left panel shows the corresponding unthresholded Z-score map. Red/orange areas indicate regions where participants exhibited stronger GBC in the LSD condition, whereas blue areas indicate regions where participants exhibited reduced GBC in the LSD condition, compared with Pla condition. The histogram above the map shows the distribution of Z-scores. (**D**) Right panel displays significant (TFCE type I error protected) areas showing increased (red) and decreased (blue) GBC in the LSD condition compared to Ket+LSD. Left panel shows the corresponding unthresholded Z-score map. Red/orange areas indicate regions where participants exhibited stronger GBC in the LSD condition, whereas blue areas indicate regions where participants exhibited reduced GBC in the LSD condition,

*Figure 1 continued on next page*

*Figure 1 continued*

compared with Ket+LSD condition. The histogram above the map shows the distribution of Z-scores. (E) Scatterplot showing a positive relationship between drug condition differences in GBC. Plotted are Z-scores for all grayordinates for the LSD>Placebo comparison (see panel C, X-axis) and LSD >Ket+LSD comparison (see panel D, Y-axis). Ellipse marks the 95% confidence interval. (F) Scatterplot showing significant negative relationship evident between averaged hyper- and hypo- connected grayordinates (based on the LSD vs. (Ket+LSD)+Pla contrast, see *Figure 1A* and inlet) across subjects (black data points) for Pla–LSD condition change scores. Grey background indicates the 95% confidence interval. (G) Retrospectively assessed (720 min after second drug administration) subjective drug-induced effects. Effects were assessed with the Five Dimension Altered States of Consciousness Questionnaire. EU: Experience of Unity; SE: Spiritual Experience; BS: Blissful State; I: Insightfulness; D: Disembodiment; ICC: Impaired Control and Cognition; A: Anxiety; CI: Complex Imagery; EI: Elementary Imagery; AVS: Audio-Visual Synesthesia; CMP: Changed Meaning of Percepts. N = 24. * indicates significant difference between LSD and Pla, and LSD and Ket+LSD drug conditions, p<0.05, Bonferroni corrected.

The online version of this article includes the following figure supplement(s) for figure 1:

**Figure supplement 1.** Effect of Ket+LSD vs. Pla on global brain connectitiy.
**Figure supplement 2.** Study Design.
**Figure supplement 3.** Quality Control (QC) measures do not correlate with mean global brain connecticity.

volume voxel in gray-matter) showing significant hyper- and hypo-connectivity for LSD compared to (Ket+LSD)+Pla conditions. Mean Fz values do not differ between Pla and Ket+LSD conditions either in hyper-connected or in hypo-connected areas. *Figure 1C* depicts the comparison between LSD and Pla conditions and *Figure 1D* the comparison between LSD and Ket+LSD conditions. Similarly to the comparison LSD vs. (Ket+LSD)+Pla shown in *Figure 1A*, LSD compared to both Pla and Ket +LSD separately induced a connectivity pattern characterized by significant (TFCE type I error protected, 10000 permutations) hyper-connectivity in predominantly sensory areas and significant hypo-connectivity in associative networks. The similarity between the LSD>Pla and LSD>Ket+LSD contrasts is corroborated by a significant positive correlation (r = 0.91, p<0.001) between the respective Z-maps (*Figure 1E*). Furthermore, only negligible differences were observed when comparing Ket +LSD and Placonditions directly (*Figure 1—figure supplement 1*). Furthermore, we tested if the directionality of LSD-induced effects on GBC (hyper-connectivity across sensory and somatomotor networks, hypo-connectivity across associative networks) are separable effects or result from functionally related systems-level perturbations. To this end, we correlated the mean connectivity strength difference between Pla and LSD in hyper-connected regions with mean connectivity strength difference in hypo-connected regions across subjects (based on the LSD vs. (Ket+LSD)+Pla contrast). There was a significant correlation between hypo- and hyper-connectivity (r = −0.90, p<0.001, *Figure 1F*) indicating that participants with the highest LSD-induced coupling within sensory and somatomotor networks also showed the strongest LSD-induced de-coupling in associative networks. This suggests that LSD-induced alterations in information flow across these networks may result from systems-level perturbations. Together, these results indicate that LSD-induced GBC alterations are predominantly attributable to its agonistic activity onto the 5-HT$_{2A}$ receptor. In line with this, a repeated-measures ANOVA (drug condition×scale) was conducted for the retrospectively administered Altered States of Consciousness (5D-ASC) questionnaire, and revealed significant main effects for drug condition (F (2, 46)=88.49, p<0.001) and scale (F (10, 230)=14.47, p<0.001), and a significant interaction of drug condition×scale (F (20, 460)=13.02, p<0.001). Bonferroni corrected simple main effect analyses showed increased ratings on all 5D-ASC scales in the LSD condition compared to Pla and Ket+LSD conditions (all p<0.05) except for the scales spiritual experience and anxiety (all p>0.20). Pla and LSD+Ket scores did not differ on any scale (all p>0.90) (*Figure 1G*).

## Influence of global signal regression on global brain connectivity following LSD administration

To investigate the influence of GSR on LSD results, we repeated the analyses presented above without GSR (i.e. the effect of drug condition on GBC shown in in *Figure 1*). The main effect of drug on GBC computed without GSR revealed significant predominantly left-hemispheric widespread differences in GBC between drug conditions (*Figure 2A*, TFCE type I error protected, 10000 permutations). *Figure 2B* shows mean Fz for each drug condition and the distribution of Fz values within voxelgrayordinates showing significant hyper- and hypo-connectivity for LSD compared to (Ket+LSD) +Pla conditions. Mean Fz values for hypo-connected grayordinates differed significantly between Pla and Ket+LSD conditions. Mean Fz values for hyper-connected grayordinates differed significantly

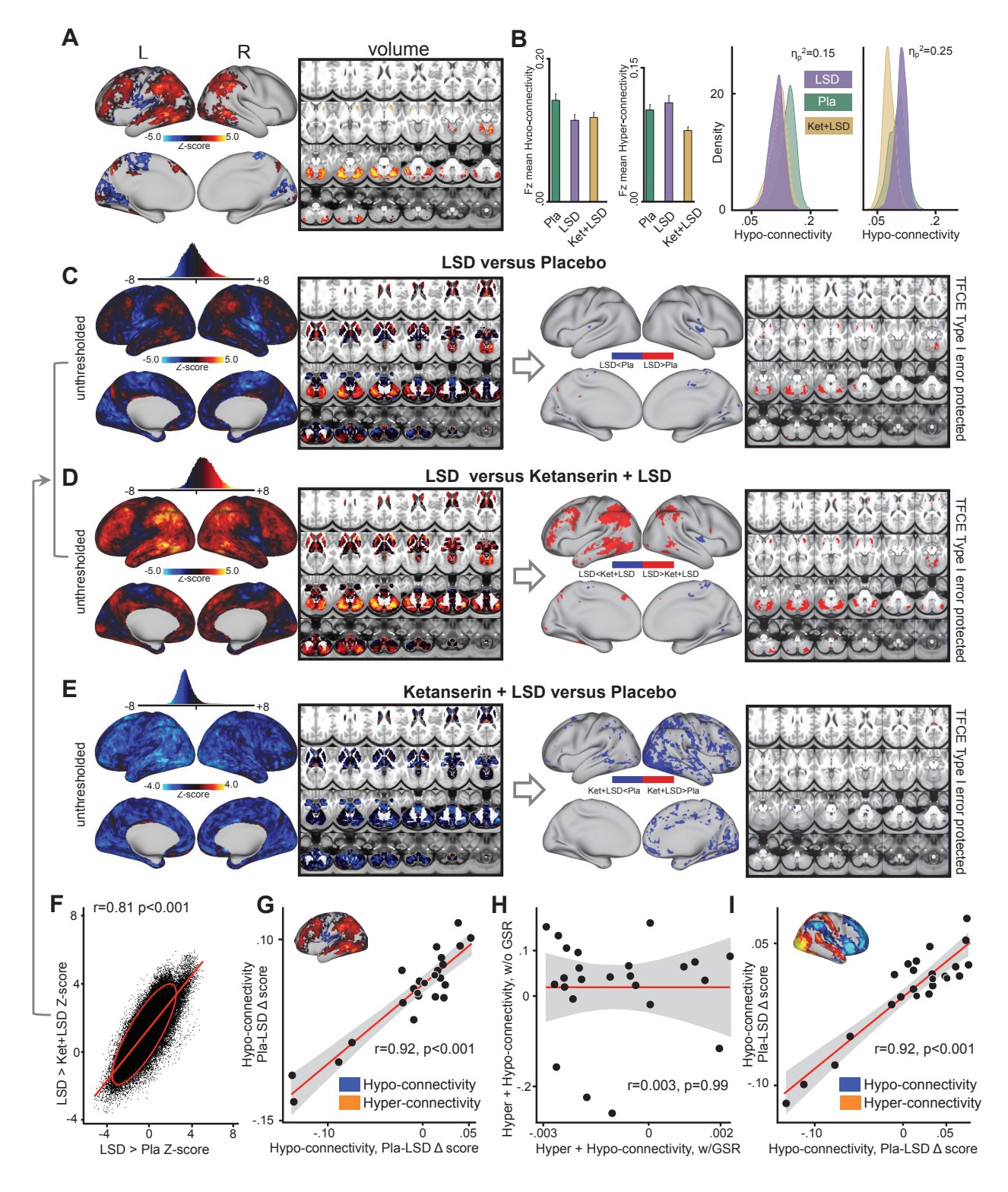

**Figure 2.** Effect of drug condition on global brain connectivity without global signal regression. (A) Z-score map for the effect of LSD condition vs. (Ket +LSD)+Pla condition within areas showing a significant main effect of drug (TFCE type I error protected). Red/orange areas indicate regions where participants exhibited stronger GBC in the LSD condition, whereas blue areas indicate regions where participants exhibited reduced GBC condition, compared with (Ket+LSD)+Pla conditions. (B) Bar plots show mean connectivity strength (Fz) values for hyper- and hypo-connected areas averaged across grayordinates showing a significant main effect of drug. Distribution plots show distribution of Fz values within grayordinates showing significant

*Figure 2 continued on next page*

Figure 2 continued

hyper- and hypo-connectivity for LSD compared to (Ket+LSD)+Pla conditions. (C) Right panel displays significant (TFCE type I error protected) areas showing increased (red) and decreased (blue) GBC in the LSD condition compared to Pla. Left panel shows the corresponding unthresholded Z-score map. Red/orange areas indicate regions where participants exhibited stronger GBC in the LSD condition, whereas blue areas indicate regions where participants exhibited reduced GBC in the LSD condition, compared with Pla condition. The histogram above the map shows the distribution of Z-scores. (D) Right panel displays significant (TFCE type I error protected) areas showing increased (red) and decreased (blue) GBC in the LSD condition compared to Ket+LSD. Left panel shows the corresponding unthresholded Z-score map. Red/orange areas indicate regions where participants exhibited stronger GBC in the LSD condition, whereas blue areas indicate regions where participants exhibited reduced GBC in the LSD condition, compared with Ket+LSD condition. The histogram above the map shows the distribution of Z-scores. (E) Right panel displays significant (TFCE type I error protected) areas showing increased (red) and decreased (blue) GBC in the Ket+LSD condition compared to Pla. Left panel shows the corresponding unthresholded Z-score map. Red/orange areas indicate regions where participants exhibited stronger GBC in the Ket+LSD condition, whereas blue areas indicate regions where participants exhibited reduced GBC in the Ket+LSD condition, compared with Pla condition. The histogram above the map shows the distribution of Z-scores. (F) Scatterplot showing a positive relationship between drug condition differences in GBC. Plotted are Z-scores for all grayordinates for the LSD>Pla comparison (see panel C, X-axis) and LSD>Ket+LSD comparison (see panel D, Y-axis). Ellipse marks the 95% confidence interval. (G) Scatterplot showing significant positive relationship evident between averaged hyper- and hypo-connected grayordinates (based on the LSD vs. (Ket+LSD)+Pla contrast without GSR, see inlet) across subjects (black data points) for Pla–LSD condition change scores without GSR. Grey background indicates the 95% confidence interval. (H) Scatterplot showing no significant relationship between Hyper +Hypo connected Fz values with GSR and Hyper+Hypo connected Fz values without GSR. Grey shading indicates the 95% confidence interval. (I) Scatterplot showing significant positive relationship evident between averaged hyper- and hypo- connected grayordinates (based on the LSD vs. (Ket +LSD)+Pla contrast with GSR, see inlet) across subjects (black data points) for Pla–LSD condition change scores without GSR. Grey background indicates the 95% confidence interval. N = 24. p<0.05, Bonferroni corrected.

between Pla and Ket+LSD, and LSD and Ket+LSD conditions. *Figure 2C* depicts the comparison between LSD and Pla conditions. Without GSR LSD induced hypo-connectivity mainly in the right insula and hyper-connectivity predominantly in the cerebellum. *Figure 2D* shows the comparison between LSD and Ket+LSD conditions with LSD-induced hypo-connectivity in the left insula and widespread predominantly left-hemispheric hyper-connectivity in the frontal and temporal cortex, the tempoparietal junction, and the cerebellum. Comparing Ket+LSD and Pla conditions revealed Ket+LSD induced hyper-connectivity predominantly in the right hemisphere (*Figure 2E*). LSD>Pla and LSD>Ket+LSD Z-maps were significantly correlated (r = 0.81, p<0.001, *Figure 2F*). To test the relationship between hyper- and hypo-connectivity when GSR was not performed, we correlated the mean Fz difference between Pla and LSD without GSR in hyper-connected regions with the mean Fz difference in hypo-connected areas (based on the LSD vs. (Ket+LSD)+Pla contrast) across subjects. There was a significant positive correlation between hyper- and hypo-connectivity (r = 0.92, p<0.001, *Figure 2G*). In contrast to the analysis performed with GSR showing a negative relationship between hyper- and hypo-connectivity change scores, the analysis without GSR indicates that participants with the highest LSD-induced hyper-connectivity showed the weakest LSD-induced de-coupling. Correlating the combined hyper- and hypo-connectivity values with GSR with those without GSR showed that these are not significantly related within subjects (r = 0.003, p=0.99, *Figure 2H*). Furthermore, we tested the consistency of the hyper/hypo relationships with and without GSR by examining the areas that survived the type I error correction following TFCE with data that has undergone GSR (*Figure 1*). Here we focused on the areas showing hyper vs. hypo effects, which we used as masks to extract values for each person prior to GSR. If GSR altered or induced the hyper/hypo effect then we would hypothesize the correlation would weaken prior to GSR. The effect was not consistent with this null hypothesis – namely that the hyper/hypo individual difference remained highly stable even without GSR (r = 0.92, p<0.001, *Figure 2I*). Put differently, this is not consistent with the hypothesis that the hyper/hypo changes are an artefact of the GSR process.

## Characterizing the directionality of LSD-induced effects on association versus sensory-somatomotor areas

We reported robust and widespread differences between GBC analyses results with and without GSR following LSD administration. This discrepancy calls into question interpretations regarding the directionality of LSD on sensory-somatomotor vs. association cortices, at least when assayed via the GBC metric. To inform this question, we conduced three additional analyses designed to investigate the influence of LSD and GSR on BOLD signal properties, which help inform and constrain GBC interpretations:

First, we investigated if the amplitude of the BOLD signal is influenced by GSR across different experimental conditions (i.e. LSD vs. Pla). Specifically, we quantified 'amplitudes' using a measure of local grayordinate-wise variance – an approach validated in our prior work in the context of clinical neuroimaging and effects of GSR in such datasets (*Yang et al., 2014*). *Figure 3—figure supplement 1* shows the change in local grayordinate-wise variance under LSD vs. Pla. The effect illustrates a very weak alteration in local variance (min/max Z = −1.54/+2.28). No areas survived whole-brain correction. This result in not consistent with the hypothesis that LSD markedly alters grayordinate-wise amplitudes/variance relative to Pla.

Second, we calculated the mean variance of the GS across all grey matter grayordinates (as opposed to local grayordinate-wise variance). We achieved this by defining the mean of all gray matter signal for a given subject based on their FreeSurfer segmentation and then computed the variance of the BOLD signal time course, averaged over all grayordinates in this global greymatter mask. Results indicate that variance of the GS does not differ significantly between conditions on average when computing the mean across all grey matter grayordinates [$F_{(2, 46)}=0.71$, $p>0.49$].

Third, we investigated the possibility that the GS itself may exhibit a shifted topography on LSD, as shown in prior work (*Yang et al., 2016b*). Specifically, the two analyses above reveal that grayordinate-wise and average GS variance do not markedly differ for LSD vs. Pla. However, the mean GS analysis above cannot address the possibility that the GS signal itself has a distinct spatial configuration following LSD administration. In other words, which areas are maximally contributing to the mean GS may not be the same after LSD administration. To investigate the possibility of a shifted topography of the GS, as shown in prior work (*Yang et al., 2016b*), we computed the beta map of the GS for each subject (see Materials and methods). This GS beta map allowed us to compare the spatial topography of GS under LSD vs. Pla conditions (*Figure 3A*). As evident from the figure, the GS beta contrast was quite robust, especially when compared to the local grayordinate-wise variance results (min/max Z = −5.73/+7.74). Critically, the map revealed a bi-directional spatial shift of the GS under LSD where associative cortices and large areas of sub-cortex showed an elevated GS contribution. In contrast, the blue areas showed a reduced GS contribution under LSD. This map correlated highly with the spatial organization of the LSD-induced changes on GBC. This is unsurprising as GBC is highly sensitive to shared brain-wide signal shifts. Put differently, a GBC measure will be sensitive to the change in the mean shared signal across the brain. If LSD is altering this mean shared signal topography, then the GBC effect should be similarly affected. To quantify this we calculated the relationship between the LSD-Pla contrast GBC map before (*Figure 3C*) and after GSR (*Figure 3B*) and the LSD-Pla contrast GS beta map. The LSD-Pla contrast GS beta map exhibited a highly significant negative correlation with the LSD-Pla contrast GBC map after GSR (r = 0.65, p<0.001), but a highly significant positive correlation before GSR (r = 0.66, p<0.001). This result provides evidence consistent with the hypothesis that LSD induces a transformation in the GS beta map itself, which is contributing the GBC effect pre/post GSR.

However, this analysis still does not inform the 'ground truth' effect of LSD on baseline connectivity – namely if LSD reduces or elevates connectivity across sensory and somatomotor versus associative cortices. There is a core limitation to the GBC metric in relation to the GS topography inherent to the way it is computed: Specifically, GBC yields the mean shared signal from a given grayordinate to all other grayordinates. This calculation is therefore affected by the shared variance across all grayordinates (i.e. the map of the GS). If this shared GS variance structure is shifted in one condition versus the next, then the GBC calculation will shift accordingly in a spatially ordered way corresponding to the GS spatial shift.

Therefore, it is not known from the GBC effects alone if LSD elevates or reduces mean connectivity in associative vs. sensory and somatomotor cortices. This interpretational challenge stems from the presented GS beta map analyses because it is not clear if GS beta map transformations on the GBC effect under LSD are primarily neural or artefactual.

To address this, we designed a complementary analysis, which yields a map that is interpretationally consistent irrespective of GS-related shifts. Here we focused on the thalamo-cortical system leveraging a well-established effect that is not affected by GSR transformations (*Yang et al., 2014*; *Anticevic et al., 2014a*; *Woodward et al., 2012*). To examine brain-wide thalamic coupling in session one we computed a seed-based map by extracting average time-series across all grayordinates in each subject's anatomically defined bilateral thalamus (via FreeSurfer segmentations). To examine between-drug differences, thalamic maps were entered into a second level analysis as done for the

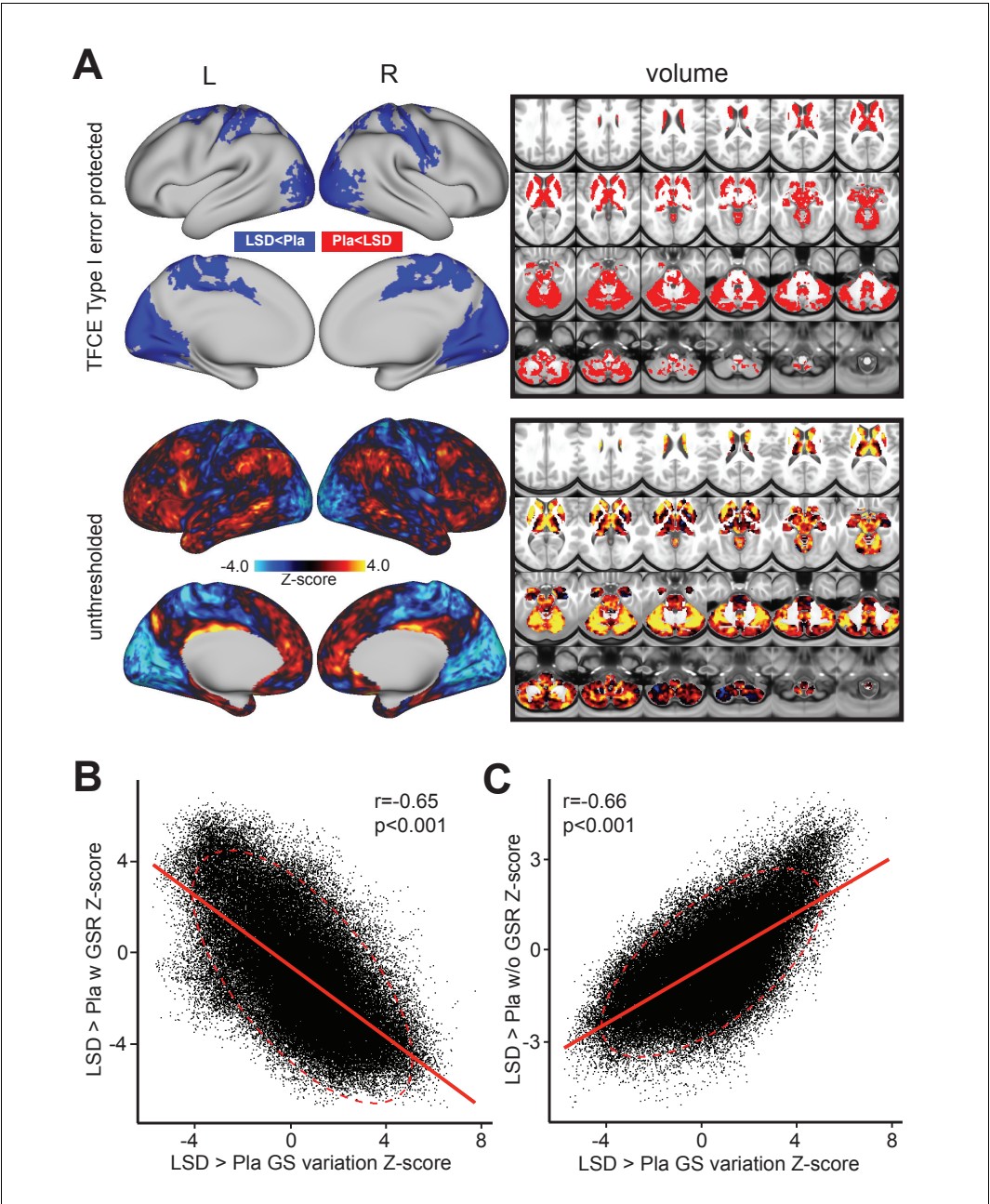

**Figure 3.** Beta map of the global signal for the LSD>Pla contrast. (**A**) The top panel displays significant (TFCE type I error protected) areas showing increased (red) and decreased (blue) covariance with the GS in the LSD condition compared to Pla. The Z map in the lower panel illustrates the unthresholded contrast between LSD vs. Pla for the GS beta map computed for each subject. Specifically, we calculated the GS (computed as mean grey matter signal) for each frame in the BOLD time course. This mean GS was then used as a regressor in a subject-specific general linear model (GLM). The resulting beta map indicates which areas are maximally co-varying with the mean GS for each subject under LSD or Pla. This 'GS beta map' was then entered into a 2nd level analysis as done for the connectivity dependent measures. This comparison tests the hypothesis that the spatial contribution to the GS is altered under LSD vs. Pla. The result shows LSD>Pla in warm colors and LSD<Pla in cool colors. (**B**) Negative correlation between Beta map of the GS for the LSD>Pla contrast and LSD>Pla GBC Z-score map after GSR and (**C**) positive correlation before GSR. N=24. The online version of this article includes the following figure supplement(s) for figure 3:

**Figure supplement 1.** Z-map of change in local voxel-wise variance under LSD vs. Pla.

GBC measures (mean thalamic connectivity maps for each condition are displayed in *Figure 4—figure supplement 1* and all contrasts in *Figure 4—figure supplement 2*). Here we used both the correlation and covariance as methods of statistical association, which we conjuncted (*Cole et al., 2016*). We did this because covariance reflects a non-normalized measure of shared BOLD signal across time (which is scale free and unaffected by variance structure) whereas correlation is inherently normalized by pooled variance. Furthermore, given that GS induces mean signal shifts (i.e. it may induce anti-correlation), we also obtained the top and bottom 10% of all thalamic connections from both correlation and covariance maps before and after GSR. This final 4-way conjunction map ensured that the resulting regions in the top/bottom ranges exhibit thalamic coupling irrespective of processing (i.e. GSR/noGSR) or statistical method (i.e. r or cov). This map was then used to calculate LSD induced effects on top 10% and bottom 10% of seed thalamic brain-wide connections. The prediction was that LSD would decrease thalamic connections that were in the top 10% (i.e. highly positive thalamic connections at baseline, which represent thalamo-associative coupling, *Figure 4A*). In turn, we predicted that LSD would elevate connections that were in the bottom 10% (i.e. very weak thalamic connections at baseline, which represent thalamo-sensory coupling, *Figure 4A*).

*Figure 4B and C* shows the difference in the average signal between drug conditions for the correlation method after GSR. Here, LSD consistently decreases coupling in associative areas and increases FC in sensory-somatomotor regions (*Figure 4A*). Critically, this effect was preserved for the LSD-(Ket+LSD) analysis (*Figure 4B*). Without GSR however, inconsistent results emerged (*Figure 4—figure supplement 2*). To reconcile the interpretation of LSD-induced directionality we leveraged the conjunction map that was robust to processing method and statistical approach. This conjunction map was used as a mask to extract the average signal across these regions in the LSD-Pla and LSD-(Ket+LSD) contrast before and after GSR in the seed-based correlation/covariance analyses as well as GBC correlation/covariance analyses. *Figure 4F* shows the difference in the average signal between these drug conditions across analyses methods (thalamic seed FC/GBC, correlation/covariance) after GSR. Here, LSD consistently decreased thalamic coupling in associative areas and increased thalamic coupling in sensory-somatomotor regions irrespective of analysis method. Importantly, the thalamic seed analyses matched GBC effects. Without GSR however (*Figure 4G*), seed thalamic coupling and GBC results were inconsistent. Furthermore, in contrast to results after GSR, LSD did not consistently decrease connections that were in the top 10% or elevate connections that were in the bottom 10% without GSR. To investigate individual differences, we computed the correlation between the top and bottom connections before and after GSR across participants (*Figure 4H*, full connectivity matrix is presented in *Figure 4—figure supplement 3*). The prediction was that individuals with biggest elevation for bottom 10% would should the biggest drop in the top 10% under LSD. Predicted negative individual differences emerged after GSR but without GSR results were not compatible with individual difference predictions based either for thalamic seed analysis or GBC. Collectively, these results are consistent with the hypothesis that, following GS cleanup, LSD reduces shared signals for association cortices but elevates shared signals for sensory and somatomotor areas across both seed-based thalamic and GBC analyses.

## Time course of subjective drug effects

To investigate the time course of subjective effects, a short version of the 5D-ASC was administered 180 min, 250 min, and 360 min after the second drug administration. A repeated-measures (drug condition×time×scale) ANOVA for the short-version 5D-ASC questionnaire revealed significant main effects for drug condtition ($F_{(2, 44)}=58.32$, $p<0.001$), time ($F_{(2, 44)}=26.61$, $p<0.001$), and scale ($F_{(4, 88)}=14.83$, $p<0.001$) and significant interactions for drug condtion×time ($F_{(4, 88)}=16.89$, $p<0.001$), treatmentdrug condition×scale ($F_{(8, 176)}=12.82$, $p<0.001$), time×scale ($F_{(8, 176)}=4.05$, $p<0.001$), and drug condition×time×scale ($F_{(16, 352)}=2.22$, $p<0.01$). Bonferroni-corrected simple main effect analyses revealed that score in the LSD treatment condition differed significantly from score in the Pla and Ket+LSD treatment conditions for the blissful state scale, disembodiment scale, elementary imagery scale, and changed meaning of percepts scale at 180 and 250 min after treatment intake (all $p<0.05$). 360 min after intake, score on the disembodiment scale and elementary imagery scale was significantly greater in the LSD treatment condition than in the Pla and Ket+LSD treatment conditions. Scores did not differ between the Pla and Ket+LSD treatment conditions for any scale at any time point (all $p>0.90$; *Figure 5*, *Figure 5—source data 1*). Test-retest reliability of these measures is high. Within each drug condition, mean scores over time correlated highly and

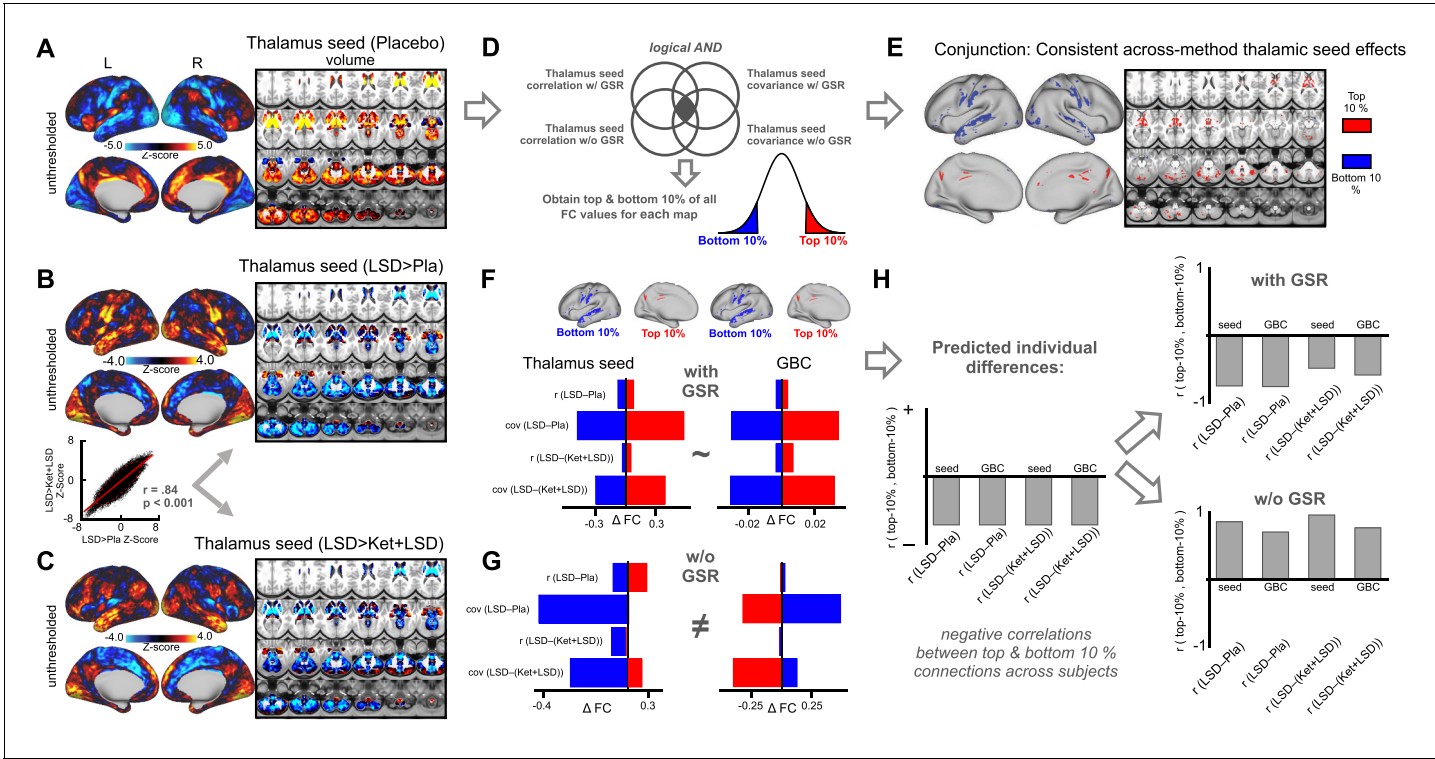

**Figure 4.** Evaluation of influence of global signal regression informed by seed-based thalamus connectivity. (**A**) The Z map illustrates the mean thalamus coupling with all grayordinates in the Pla condition. Warm colors indicate positive connections with the thalamus. Cool colors indicate negative connections with the thalamus. (**B**) The Z-map shows thalamus seed-based connectivity for the LSD>Pla contrast. Warm colors indicate increased thalamus connectivity in the LSD condition. Cool colors indicate decreased thalamus connectivity in the LSD condition. The scatterplot shows the correlation between Z-maps displayed in D and E. (**C**) The Z-map shows thalamus seed-based connectivity for the LSD>Ket+LSD contrast. Warm colors indicate increased thalamus connectivity in the LSD condition. Cool colors indicate decreased thalamus connectivity in the LSD condition. (**D**) Schematic illustrating the conjunction analysis. The top/bottom 10% of all connections were extracted from the mean connectivity (correlation and covariance) maps in the Pla condition before and after GSR and used to compute a conjunction map providing the strongest and weakest thalamic connections irrespective of analysis method. (**E**) Result of conjunction analysis used as mask to extract values in the following analyses. (**F**) Mean differences between drug conditions within top and bottom regions revealed by the conjunction analysis before GSR for thalamus seed connectivity (correlation and covariance) and GBC (correlation and covariance). (**G**) Mean differences between drug conditions within top and bottom regions revealed by the conjunction analysis after GSR for thalamus seed connectivity (correlation and covariance) and GBC (correlation and covariance). (**H**) The right bar graph illustrates that correlation coefficients between top and bottom area vales across participants are expected to be negative. The upper right panel shows the correlation coefficients between top and bottom connections revealed by the conjunction analysis after GSR. The lower right panel shows the correlation coefficients between top and bottom connections revealed by the conjunction analysis before GSR. r: correlation; cov: covariance, N = 24.

The online version of this article includes the following figure supplement(s) for figure 4:

**Figure supplement 1.** Thalamus seed-based connectivity.

**Figure supplement 2.** Thalamus seed-based connectivity contrast maps.

**Figure supplement 3.** Across-subject correlation matrix between top and bottom connections and different analysis methods.

significantly (all Pearson's r > 0.40, max r = 0.99). *Figure 5—figure supplement 1* shows the correlation coefficients between scores on the five subscales in the LSD condition at the three time points.

## Session impacts global brain connectivity in Ketanserin+LSD condition

To investigate the potentially distinct temporal phases of LSD pharmacology (*Marona-Lewicka et al., 2005*; *Marona-Lewicka and Nichols, 2007*), two resting-state scans were conducted on each test day: 75 min (session 1) and 300 min (session 2) after the second drug administration. No significant differences in GBC were observed when comparing session 1 and 2 within the Pla and the LSD condition (*Figure 6—figure supplement 1*). Within the Ket+LSD condition, participants showed significant decreases in GBC in session two compared to session one predominantly in

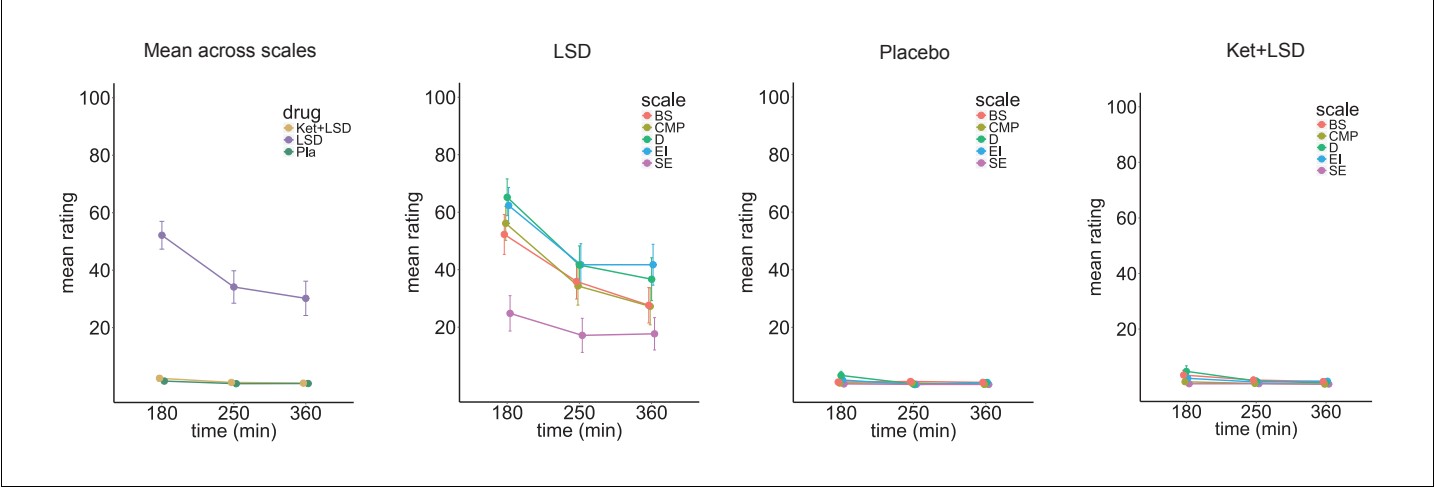

**Figure 5.** Time course of subjective drug effects. Five Dimension Altered States of Consciousness Questionnaire short version scores assessed at 180, 250, and 300 min after second drug administration for the means across scales, and scale scores for Pla, LSD, and Ket+LSD conditions. Scores are expressed as percent of the scale maximum. Data are expressed as means ± the standard error of the mean (SEM). BS: Blissful State; CMP: Changed Meaning of Percepts; D: Disembodiment; EI: Elementary Imagery; SE: Spiritual Experience. N = 23.

The online version of this article includes the following source data and figure supplement(s) for figure 5:

**Source data 1.** Five Dimension Altered States of Consciousness Questionnaire short version.

**Figure supplement 1.** Across-subject correlation.

occipital areas. Increases in GBC in session two were found in cortical regions such as the anterior and posterior cingulate cortex, and the temporoparietal junction, as well as subcortical structures including the thalamus and the basal ganglia (*Figure 6A*). *Figure 6B* shows the significant (p<0.05) difference in mean Fz between session 1 and session two in hyper- and hypo-connected areas and the distribution of Fz values for grayordinates within hyper- and hypo-connected areas for both sessions (hyper- and hypo-connected areas are derived from the LSD vs. (Ket+LSD)+Pla contrast, see *Figure 1A*).

We next specifically investigated mean Fz (with and without GSR) for all drug conditions and sessions for grayordinates within seven functionally-defined networks using parcellations derived by *Yeo et al. (2011)*, *Buckner et al. (2011)* and *Choi et al. (2012)* (*Figure 7*). This parcellation contains both sensory (visual and somatomotor) and associative (dorsal attention, ventral attention, limbic, frontoparietal control, and default mode) networks. Repeated-measures ANOVAs revealed significant main effects for drug condition for all networks (all p<0.05) except for the dorsal attention network when including GSR, with the LSD condition differing significantly from both, Pla and Ket +LSD conditions (all p<0.05, Bonferroni corrected), except for the somatomotor network, where LSD differed significantly only from Ket+LSD. Pla and Ket+LSD conditions did not differ significantly in any network. Without GSR, main effects for drug were found in the frontoparietal control network (F (2, 46)=4.09, p<0.03) with significantly lower values in the Ket+LSD condition than in both, the LSD and Pla condition, and dorsal attention network (F (2, 46)=3.86, p<0.04) with significantly lower values in the Ket+LSD condition than in the Pla condition.

## Global brain connectivity in somatomotor network correlates with subjective effects

To evaluate the relationship between LSD-induced changes in GBC in functional networks and subjective LSD-induced effects, Fz mean connectivity change (LSD–Pla condition, session 2, with GSR) in the seven functional networks (see *Figure 7*) was correlated with the mean 5D-ASC short version score at 250 mins (assessment closest in time to resting-state data collection, see *Figure 1—figure supplement 2* and *Figure 5*). Correlating measures at session two allows high stability in LSD-induced effects. Bonferroni corrected correlations showed a significant relationship between the change in Fz connectivity in the somatomotor network and subjective LSD-induced effects (r = 0.81, p<0.001, Bonferroni corrected, *Figure 8A* and *Figure 8B*). Correlations between mean 5D-ASC

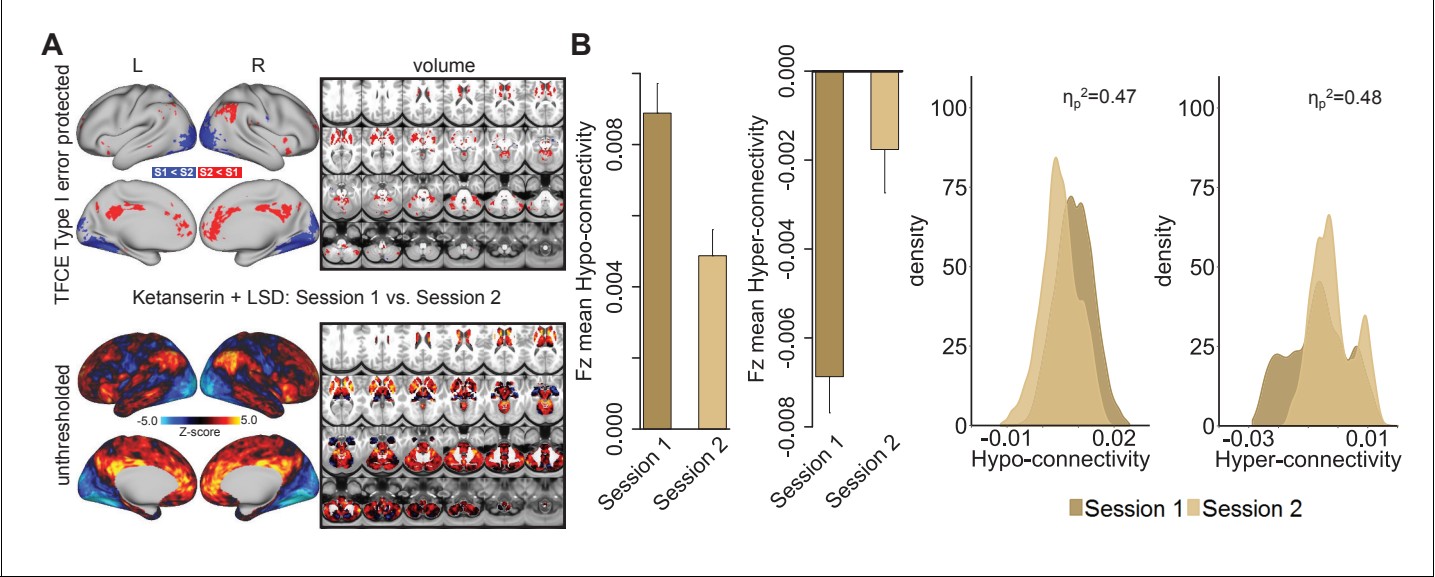

**Figure 6.** Effect of session on global brain connectivity in the Ketanserin+LSD condition. (**A**) Top panel displays significant (TFCE type I error protected) areas showing increased (red) and decreased (blue) GBC in session 1 (75 minutes after second drug administration) compared to session 2 (300 minutes after second drug administration). Lower panel shows the corresponding unthresholded Z-score map. Red/orange areas indicate regions where participants exhibited stronger GBC in session 1, whereas blue areas indicate regions where participants exhibited reduced GBC in session 2. (**B**) Bar plots show mean connectivity strength (Fz) values for hyper- and hypo-connected areas (significant for the LSD vs. (Ket+LSD)+Pla contrast) for session 1 and session two in the Ket+LSD condition. Distribution plots show distribution of connectivity strength (Fz) values within grayordinates showing hyper- and hypo-connectivity (significant for the LSD vs. (Ket+LSD)+Pla contrast) for session 1 and session two in the Ket+LSD condition. N = 24.

The online version of this article includes the following figure supplement(s) for figure 6:

**Figure supplement 1.** Effect of session on global brain connectivity.

score and Fz connectivity in the other six networks and did not reveal significant relationships (all p>0.16, Bonferroni corrected). To further investigate the contribution of specific LSD-induced symptoms to the relationship with somatomotor network Fz connectivity, we calculated the correlation between Fz mean connectivity change in the somatomotor network with each 5D-ASC short version scale separately. All five scale scores (blissful state, disembodiment, changed meaning of percepts, elementary imagery, spiritual experience) were significantly correlated with Fz mean connectivity change in the somatomotor network (all p<0.05, Bonferroni corrected, *Figure 7C–G*), indicating that the relationship between somatomotor network Fz connectivity and subjective effects was not driven by a specific LSD-induced symptom alone. The five scale scores were moderately to strongly correlated with each other (r = 0.39 – 0.82). Correlating mean Fz connectivity changes without GSR in the seven functional networks with subjective effects did not reveal any significant result (all p>0.3, unc).

## GBC maps with GSR correlate predominantly with HTR2A and HTR7 cortical gene expression maps

LSD stimulates not only 5-HT$_{2A}$ receptors but also 5-HT$_{2C}$, $_{1A/B}$, $_6$, and $_7$ and dopamine D2 and D1 receptors. These receptors are differentially expressed across the cortex. To further investigate LSD's receptor pharmacology, we tested the correlation between unthresholded Z-score map for LSD condition vs. (Ket+LSD)+Pla condition with and without GSR and six available receptor gene expression maps of interest (DRD1, DRD2, HTR1A, HTR2A, HTR2C, and HTR7) derived from the Allen Human Brain Atlas (*Burt et al., 2018*; *Hawrylycz et al., 2012*). *Figure 9A* shows the average GBC Z-score with and without GSR and the mean gene expression of the genes of interest within the seven functionally-defined networks using parcellations derived by *Yeo et al. (2011)*, *Buckner et al. (2011)* and *Choi et al. (2012)* (see also *Figure 7*), indicating that gene expression is distinct by network. Next, we investigated whether there is a common pattern of distribution between the six gene expression maps. Correlation analyses showed that the expression of the main

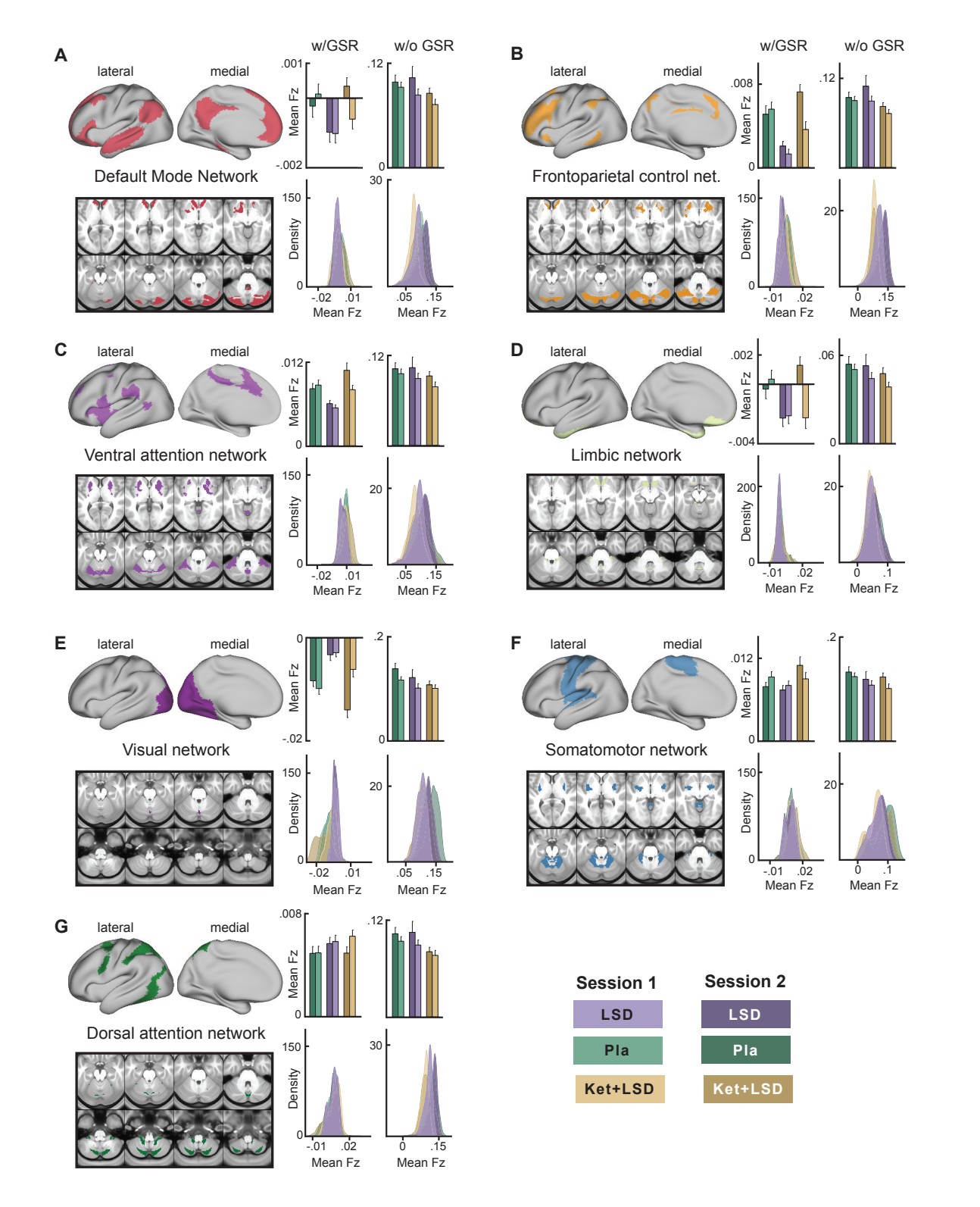

**Figure 7.** Effect of drug condition, session, and globals signal regression on global brain connectivity in functional networks (A–G). Brain maps illustrate lateral, medial, and subcortical view of functional networks. Bar plots show mean connectivity strength (Fz) values for grayordinateswithin functional networks for Pla, LSD, and Ket+LSD conditions, for session 1 and session two respectively, as well as with and without GSR. Distribution plots show

*Figure 7 continued on next page*

*Figure 7 continued*

distribution of connectivity strength (Fz) values for grayordinates within functional networks for Pla, LSD, and Ket+LSD conditions, for session 1 and session 2 respectively, as well as with and without GSR. N = 24.

gene of interest (HTR2A) is highly negatively correlated with the expression of HTR7 (r = −0.68, p<0.001, Bonferroni corrected, *Figure 9B*). *Figure 9C* illustrates the cortical distribution of HTR2A gene expression. This HTR2A cortical gene expression map is highly correlated with the unthresholded GBC Z-score map for the LSD condition vs. (Ket+LSD)+Pla condition with GSR (r = 0.50, p<0.001), and higher than all other candidate serotonin receptor genes. While the correlation between the unthresholded GBC Z-score map without GSR and the HTR2A cortical gene expression map also reached significance (r = 0.18, p<0.001), this correlation was significantly weaker than between the Z-score map with GSR and the HTR2A gene expression map (p<0.05, Bonferroni corrected, *Figure 9F*). Taking into account all available gene expression maps the correlation between the Z-score map with GSR and the HTR2A gene expression map was higher that 95.9% of all possible correlations. The GBC Z-score map with GSR and the HTR7 gene expression map was lower than 99.8% of all possible correlations, indicating a strong negative relationship (r = −0.63, p<0.001, *Figure 9D*). The correlation between both HTR2A and HTR7 with the GBC Z-Score map is not surprising considering the strong negative correlation between HTR2A and HTR7 gene expression maps (*Figure 9B*). Lastly, *Figure 9F* illustrates that correlation coefficients between gene expression maps and GBC Z-score maps were significantly stronger with GSR (all p<0.05, Bonferoni corrected),

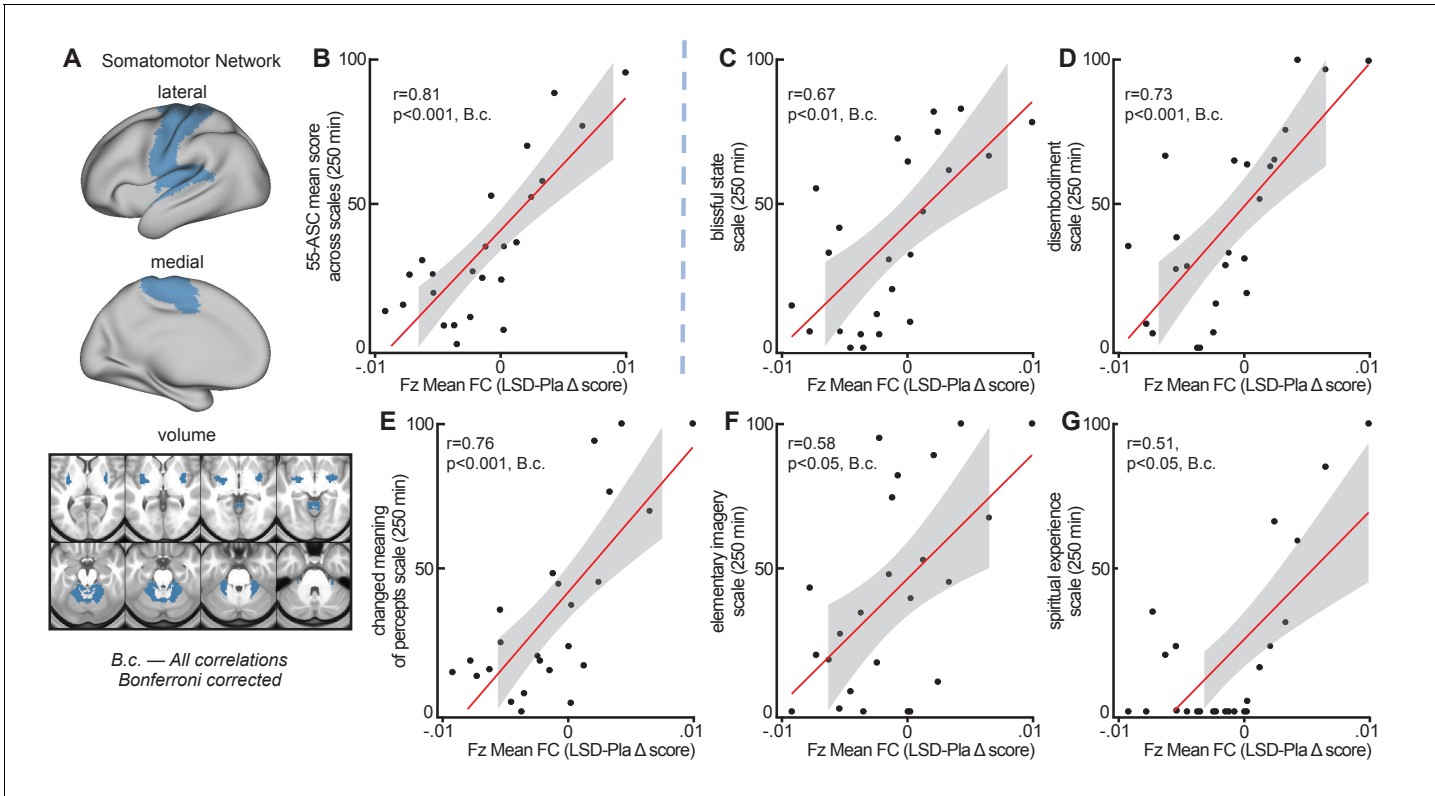

**Figure 8.** Correlation between global crain connectivity and subjective effects. (A) The brain map illustrates lateral, medial, and subcortical view of the somatomotor network. (B) The scatterplot shows the significant positive correlation between Fz mean connectivity change (LSD–Pla condition, session 2, with GSR) in the somatomotor network and the mean 5-DASC short version score at 250 mins. (C-G) Scatterplots show the positive correlation between Fz mean connectivity change (LSD–Pla condition, session 2, with GSR) in the somatomotor network and the five subscales of the 5-DASC short version score at 250 mins: blissful state, disembodiment, changed meaning of percepts, elementary imagery, spiritual experience. B.c.: Bonferroni corrected. Grey background indicates the 95% confidence interval. N = 24.

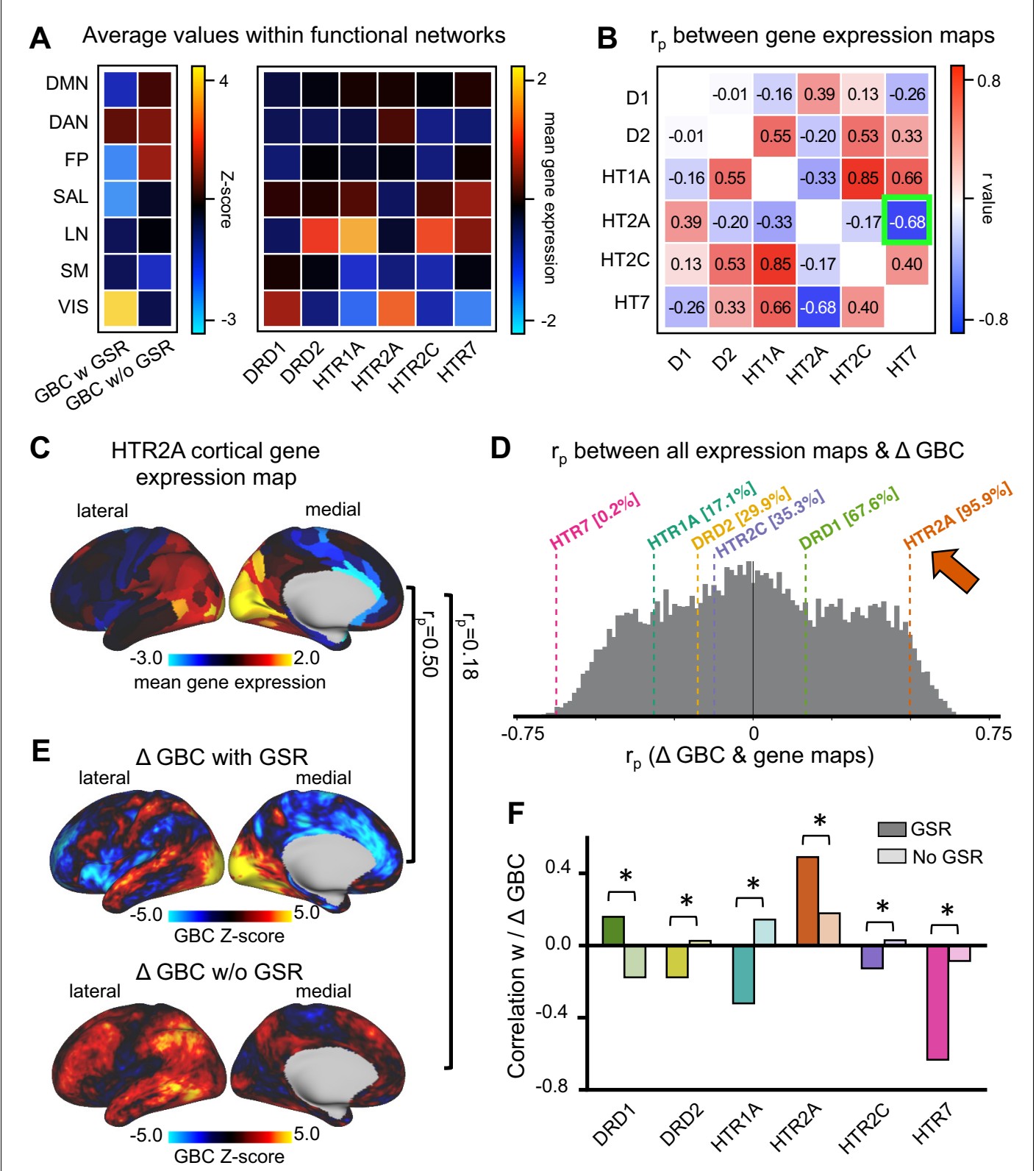

**Figure 9.** Correlation between global brain connectivity and cortical gene expression maps. (A) The top left panel shows the average GBC Z-score (LSD condition vs. (Ket+LSD)+Pla condition) with and without GSR and mean gene expression value within functional networks. (B) The top right panel shows the correlation (Pearson's *r*) between the gene expression maps, highlighting (green) the negative correlation between the expression maps of HTR2A and HTR7. (C) The brain map illustrates the cortical expression levels (Z-score) of HTR2A in the left hemisphere (lateral and medial view). (D) The

*Figure 9 continued on next page*

*Figure 9 continued*

histogram depicts the correlation between all gene expression maps and the unthresholded Z-score map for LSD condition vs. (Ket+LSD)+Pla condition with GSR. The colored lines highlight the gene expression maps of interest. (**E**) Unthresholded Z-score map for LSD condition vs. (Ket+LSD)+Pla with (top) and without (bottom) GSR. Red/orange areas indicate regions where participants exhibited stronger GBC in the LSD condition, whereas blue areas indicate regions where participants exhibited reduced GBC in the LSD condition, compared with (Ket+LSD)+Pla. $r_p$ values are the respective correlation coefficients between Z-score maps and HTR2A gene expression map. (**F**) The bar graph shows the correlation coefficients (Pearson's *r*) between each gene expression map of interest and Z-score maps for LSD condition vs. (Ket+LSD)+Pla condition with and without GSR. * indicates significant difference between correlations between Z-score map with and without GSR and gene expression map, p<0.05, Bonferroni corrected. The online version of this article includes the following figure supplement(s) for figure 9:

**Figure supplement 1.** Schematic illustrating the process of generating cortical gene expression maps from the Allen Human Brain Atlas (AHBA).

---

except for DRD1 expression map, where the absolute value of the correlation coefficient increased when correlated with the Z-score map without GSR.

## Discussion

Interest in the potential clinical effects of psychedelics is boosted by positive preliminary reports on the safety and tolerability in healthy participants as well as patient populations (*Carhart-Harris et al., 2016a*; *Gasser et al., 2014*; *Carhart-Harris and Goodwin, 2017*). However, the underlying neuropharmacology psychedelics is scarcely investigated in humans. The current study closes major knowledge gaps in the area by characterizing the effects of the prototypical psychedelic by showing that i) LSD increases GBC across sensory and somatomotor functional networks and reduces GBC in associative networks, which is sensitive to GS removal; ii) time-dependent effects are only found in the interaction with katenserin; iii) GBC in the somatomotor network was associated with subjective effects; iv) LSD-induced effects on GBC and subjective symptoms are linked to the pharmacology of the 5-HT$_{2A}$ receptor; v) innovative gene expression analyses across cortex reveal for the first time a correspondence between specific spatial gene expression patterns and in-vivo pharmacological effects in humans.

### LSD increases GBC across sensory and somatomotor functional networks and reduces GBC in associative networks

We show across conditions that LSD induces hyper-connectivity predominantly in sensory and somatomotor areas (i.e. the occipital cortex, the superior temporal gyrus, and the postcentral gyrus). LSD-induced hypo-connectivity was observed across subcortical areas (with the exception of the amygdala and sensory thalamus) as well as cortical areas associated with associative networks (i.e. the medial and lateral prefrontal cortex, the cingulum, the insula, and the temporoparietal junction) (*Figure 1*). These results are consistent with the hypothesis that LSD induces a de-synchronization of associative networks whereas sensory and somatomotor areas exhibit elevated brain-wide shared signal (*Anticevic et al., 2014b*). This is in line with previous seed-based studies reporting increased V1 resting-state connectivity with the rest of the brain after LSD administration (*Carhart-Harris et al., 2016b*). Additionally, decreases of connectivity within the DMN were reported after ayahuasca, a tea containing the hallucinogenic 5-HT$_{2A}$ receptor agonist N,N-Dimethyltryptamine, intake (*Palhano-Fontes et al., 2015*). Furthermore, desynchronization within the DMN was reported after psilocybin infusion measured with magnetoencephalography (*Muthukumaraswamy et al., 2013*). As noted, while subcortical areas predominantly show hypo-connectivity under LSD there were key exceptions: the amygdala exhibited brain-wide hyper-connectivity under LSD. Amygdala neurons abundantly express 5-HT$_{2A}$ receptors, and alterations in amygdala activity and connectivity have been hypothesized to be important for potential beneficial clinical effects of psychedelics (*Rainnie, 1999*; *Kraehenmann et al., 2016*). Furthermore, results showed that participants with the highest LSD-induced coupling within sensory and somatomotor networks also showed the strongest LSD-induced de-coupling in associative networks. This suggests that LSD-induced alterations in information flow across these networks probably results from linked systems-level perturbations, as opposed to being due to dissociable mechanisms across subjects. This pattern of hyper-and hypo-connectivity may underlie the psychedelic state, suggesting increased processing of sensory information which is not counterbalanced by associative network integrity. Consequently,

this may result in an altered state of consciousness whereby internal and external sensory computations are not integrated, leading to psychedelic symptoms.

A virtually identical pattern of hyper-connectivity in sensory networks and hypo-connectivity in associative networks is revealed when contrasting LSD effects with the condition where LSD is blocked by pre-administration of Ket. This brain-wide net effect of LSD was virtually indistinguishable from LSD vs. Pla condition. Put differently, pre-treatment of LSD with Ketanserin induced only negligible changes compared to LSD vs. Pla, indicating that Ket blocked virtually all LSD-induced alterations in GBC. Ketanserin has high antagonistic properties particularly on the 5-HT$_{2A}$ receptor. Thus, these results indicate that LSD-induced alterations in neural and behavioral effects are highly dependent on stimulation of the 5-HT$_{2A}$ receptor (*Leysen et al., 1982*). This is in line with data on subjective LSD-induced effects which were normalized by Ket pretreatment (*Figure 1G*).

Collectively, these data extend previous studies by revealing that the described pattern of brain-wide dysconnectivity may be directly attributable to stimulation of the 5-HT$_{2A}$ receptor. Specifically, the effect can be characterized by brain-wide integration of sensory networks and dis-integration of associative networks, which presumably underlie LSD-induced altered state of consciousness. Furthermore, these data highlight the importance of the 5-HT$_{2A}$ receptor system in LSD-induced neural and behavioral alterations.

## LSD's effect on GBC is sensitive to GS removal

One small study by Tagliazucchi et al. has previously investigated the effects of intravenously administered LSD and reports that association cortices (partially overlapping with the default-mode, salience, and frontoparietal attention networks) and the thalamus showed increased GBC under LSD (*Tagliazucchi et al., 2016*). These results are partially contradictory to the data presented in *Figure 1*. Importantly, this previous study did not quantify the influence of the GS, which likely contains a complex mixture of non-neuronal artefacts (e.g. respiration, which may be increased under LSD [*Power et al., 2017*; *Glasser et al., 2017*]). Such artefacts can induce spuriously high statistical association across the brain (*Yang et al., 2014*; *Coyle, 2006*). Due to these discrepancies, we studied the data as a function of GS regression to inform how this methodological step affects results (*Tagliazucchi et al., 2016*). These analysis support the observations by *Tagliazucchi et al. (2016)* when GS was not considered: results showed increased GBC in fronto-parietal, temporal, and subcortical areas (*Figure 2*). The analysis without GSR showed a positive correlation between hyper- and hypo-connectivity change scores, indicating that participants with the highest LSD-induced hyper-connectivity showed the weakest LSD-induced de-coupling. This was in contrast to results after GS removal. Furthermore, connectivity values with GSR and those without GSR were not significantly correlated within subjects (*Figure 2*). Notably, without GS regression, LSD-Pla differences were not observed when examining seven functionally pre-defined networks (*Figure 7*). These data suggest that GS related effects cannot be explained by a mean-shift in connectivity values on average, but instead may reflect a process within each subject.

One hypothesis is that GSR statistically attenuates non-neural arts and therefore provides a method to better isolate functional networks in pharmacological resting-state connectivity studies (*Yang et al., 2014*; *Coyle, 2006*). This interpretation is consistent with the absence of a neural-symptom correlation without GSR. Finally, spatial correlations with gene expression maps (discussed below) were notably attenuated for without GSR. That said, this dataset is not well-suited for drawing conclusions about GSR suitability for pharmacological neuroimaging. In fact, it can be argued that results are more replicable across session 1 and 2 without GSR. However, the statistical phenomenon of 'artefactual' replication is not surprising, if one considers that GSR is designed to attenuate sources of spatially pervasive structured artefacts which may persist across sessions; f.e. elevated respiratory artefacts). Put differently, there is a key nuance between 'noise' and 'artefact'. Pure unstructured noise can be signal-averaged out and would not yield a consistent 'artefactual' effect. In contrast, 'structured' artefact represents a signal that can induce the same spurious effect multiple times (*Power et al., 2018*; *Glasser et al., 2018*). Therefore, if a structured artefact is large across both measurements (session 1 and 2), then this artefact will spuriously drive the effect and will be replicable. Experiments that manipulate variables such as breathing rate and vigilance will be key to fully characterize the effects of GSR on pharmacological neuroimaging data and help separate neuropharmacological effects associated with 'global artefacts' versus those affecting 'global neural signal' (*Glasser et al., 2018*). Furthermore, there are open knowledge gaps regarding LSD's effects

on neurovascular coupling and the hemodynamic response function properties. This pitfall needs to be addressed in experiments incorporating measures specifically designed to investigate changes in hemodynamic coupling. Here animal studies, which offer the possibility to combine neuronal recordings with simultaneous measurement of hemodynamics, will be critical to help interpret LSD effects inhumans.

## Considering the directionality of LSD's neural effects

As expected, the GS analyses indicate that a GBC metric is highly sensitive to global shared signal, which is altered under LSD. This raises the question regarding the direction of the LSD-induced effects on association vs. sensory-somatomotor areas. To inform the directionality of LSD modulations of GBC we completed an additional analyses based on seed-based thalamic functional connectivity, which yielded a map that was robust to GS transformations. The reason for this phenomenon is that the thalamus exhibits strong bi-directional brain-wide shared signal. Furthermore, we constructed a conjunction measure that identified baseline (i.e. Pla) thalamic FC that was interpretationally consistent irrespective of GS-related shifts (*Figure 4*). This seed-based conjunction analysis revealed that LSD-induced changes were consistent after GSR and comparable to GBC effects. Without GS removal, neither the thalamic nor GBC effects converged across metrics. This observation, however, does not rule out the possibility that GS removal in fact attenuates signal components that are neuronal in origin and may be relevant to important LSD-induced properties (*Tagliazucchi et al., 2016*). Careful manipulation and measurement of respiratory-related artefacts during pharmacological fMRI is needed to disambiguate the amount of GS variance that relates to neuronal vs. artefactual LSD effects.

## Time-dependent effects of LSD

Animal studies suggest distinct temporal phases of LSD pharmacology (*Marona-Lewicka et al., 2005*; *Marona-Lewicka and Nichols, 2007*). Therefore, we investigated the time-dependent effects of LSD on subjective effects as well as on GBC. As shown in *Figure 5*, subjective effects were highest 180 min after LSD administration and decreased in intensity 250 and 360 min after administration as expected (*Passie et al., 2008*; *Schmid et al., 2015*). No differences were found between Pla and Ket+LSD conditions at any time point, with subjective effects in both conditions being very low in intensity (<4.7%). This shows that Ket blocked subjective LSD-induced effects over the whole time course, indicating that subjective effects are most likely attributable to 5-HT$_{2A}$ receptor stimulation. To investigate the time course of LSD-induced effects on GBC, two resting-state scans were analyzed, conducted 75 min (session 1) and 300 min (session 2) after the second drug administration. While no significant differences were observed when comparing session 1 and 2 within the Pla and the LSD condition (*Figure 6—figure supplement 1*), participants showed significant changes in GBC in session two compared to session one in the Ket+LSD condition. Taken together with time-dependent results observed in the functionally defined networks (*Figure 6*), the blocking effect of ketanserin is particularly evident in the session one across all networks. Specifically, Ket not only blocks LSD effects in session one but also augments the effects seen in Pla, indicating the opposite mechanism of action from that seen by LSD – namely 5-HT$_{2A}$ receptor antagonism (*Leysen et al., 1982*; *Kometer et al., 2013*). On the other hand, it seems possible that there exist two distinct pharmacological time phases as described in animal studies. The first phase may be modulated by 5-HT$_{2A}$ receptor activation and the second phase possibly by D2 receptor activation, as suggested by preclinical work (*Marona-Lewicka et al., 2005*; *Marona-Lewicka and Nichols, 2007*). This hypothesis of time-dependent complex receptor pharmacology awaits further testing. First, studies are needed to investigate the effect of Ket alone on GBC to verify the preferential effects of 5-HT$_{2A}$ antagonism. Second, studies using pre-treatment of LSD by antagonists on receptors other than 5-HT$_{2A}$ are needed to determine if the second phase is indeed modulated by another receptor system. Lastly, indications of different pharmacological phases are not evident from subjective drug effects which remain completely blocked by Ket. Studies using higher doses of LSD are therefore needed to investigate if the potential effect of LSD's action on other receptors becomes more pronounced and therefore subjectively accessible.

## LSD-induced alterations in GBC in the somatomotor network are associated with subjective effects

The LSD-induced change in connectivity in the somatomotor network correlated significantly with general and specific subjective LSD-induced effects (mean across all scales, blissful state, disembodiment, changed meaning of percepts, elementary imagery, spiritual experience). Participants with increased connectivity in the somatomotor network also showed higher subjective effects. On average, the change in connectivity between the LSD and Pla condition in the somatomotor network was not significant. However, this is likely explained by the heterogeneous connectivity changes within this network: while the pre- and postcentral gyrus predominantly showed increases in GBC, medial areas were hypo-connected. Connectivity changes in other functional networks were not significantly correlated with subjective effects. This points to the importance somatomotor network brain regions and their connectivity with the rest of the brain for psychedelic experiences. This is in line with previous results obtained from task-related data showing that the supplementary motor area is associated with LSD-induced alterations in meaning and personal relevance processing (*Preller et al., 2017*). These results also support broader theories of consciousness emphasizing the importance of the sensorimotor system for the perception of presence and agency, and therefore a sense of self (e.g., interoceptive predictive coding model of conscious presence (*Seth et al., 2011*), comparator model (*Frith et al., 2000*; *Allen et al., 2016*; *Blanke and Metzinger, 2009*). Furthermore, alterations in sensorimotor gating have been suggested to underlie psychedelic experiences (*Quednow et al., 2012*; *Ludewig et al., 2003*). Somatomotor system activity and connectivity has also been implicated in the pathophysiology of schizophrenia (*Anticevic et al., 2014a*), an illness characterized by delusions and alterations in the sense of self, potentially arising from alterations in sensorimotor gating deficits in an inferential mechanism that allows distinguishing whether or not a sensory event has been self-produced (*Synofzik et al., 2010*). The current results corroborate and extend these previous findings by showing that somatomotor network connectivity is also closely associated with an LSD-induced psychedelic state.

## LSD-induced alterations in GBC correlate with HTR2A and HTR7 cortical gene expression

To further investigate LSD's receptor pharmacology we specifically used the threshold-free Z-score map of LSD effects relative to Ket blockade and Pla. The logic here is that such a map may reflect Ket-specific contributions to LSD blockade, which is hypothesized to involve the $5\text{-}HT_{2A}$ receptor. This map was then correlated with gene expression maps of receptors that may be stimulated by LSD (*Nichols, 2004*). LSD-induced changes in functional connectivity after GSR exhibited strong positive relationships with HTR2A expression (higher than 95.9% of all possible gene expression correlations, *Figure 9*). These results show that LSD-induced changes in GBC quantitatively match the spatial expression profile of genes coding for the $5\text{-}HT_{2A}$ receptor, supporting the central role of this receptor system in LSD's neuronal and subjective effects. LSD-induced changes in functional connectivity were also highly negatively correlated with HTR7 gene expression (lower than 99.8% of all possible gene expression correlations, *Figure 9*). This can be explained by the highly anti-correlated expression of these two genes (*Figure 9*). However, it is also possible that the $5\text{-}HT_7$ receptor functionally contributes to LSD-induced effects. In contrast to its agonistic properties on the $5\text{-}HT_{2A}$ receptor, LSD has been reported to be an antagonist in the $5\text{-}HT_7$ receptor (*Wacker et al., 2013*). Since previous studies have shown that $5\text{-}HT_7$ receptor antagonists have anti-psychotic potential (*Waters et al., 2012*; *Abbas et al., 2009*), it seems very unlikely that LSD's effects have a strong and appreciable contribution on the $5\text{-}HT_7$ receptor. However, future studies should examine $5\text{-}HT_7$ receptor pharmacology more carefully as they may reveal a role of this receptor system in pro-cognitive effects that contrast those of LSD. While the current results strongly implicate the involvement of the $5\text{-}HT_{2A}$ receptor in LSD-induced effects, it must be noted that no further conclusions can be drawn regarding the functional contribution of other receptors agonized or antagonized by LSD. This limitation needs further investigation in future studies by blocking serotonin and dopamine receptors involved in the pharmacology of LSD beyond the $5\text{-}HT_{2A}$ receptor. Furthermore, the contribution of these receptors to the effects of different doses of LSD still need to be studied. Finally, we show that the spatial match between gene expression maps and GBC maps is significantly improved after GSR, even though correlation coefficients in particular for DRD1, DRD2,

HTR1A, and HTR2C remain moderate. These results also highlight the validity of this approach as a general method for relating spatial gene expression profiles to neuropharmacological manipulations in humans. An important next step allowing further methodological validation is comparing LSD-induced alterations in GBC with receptor maps provided by Positron Emission Tomography (PET) (*Saulin et al., 2012*; *Ettrup et al., 2016*; *Ettrup et al., 2014*), preferably using MR scanners that are both MR and PET compliant allowing for cross-validation across BOLD and PET modalities within the same person.

## Conclusion

In summary, the current results close major knowledge-gaps regarding the neurobiology and neuro-pharmacology of LSD. First, we show that LSD induces widespread alterations of GBC in cortical and subcortical brain areas, characterized by a synchronization of sensory and somatomotor functional networks and dis-integration of associative networks. We show that this effect is sensitive to GSR, which has important implications for future pharmacological resting-state studies. Second, we investigated the receptor-pharmacology of LSD, showing that the 5-HT$_{2A}$ receptor plays a critical role in subjective and neuronal LSD-induced effects. However, analyzing the time course of LSD-induced alterations in functional connectivity, it seems likely that at a later phase, modulation by receptors other than the 5-HT$_{2A}$ receptor is involved. The comparison of LSD-induced effects on functional connectivity and receptor-gene expression maps underscores the interpretations of 5-HT$_{2A}$ pharmacology and points to potentially impactful studies on 5-HT$_7$ receptor pharmacology. Lastly, in line with various theories of consciousness we showed that the somatomotor system in particular is related to LSD-induced psychedelic effects. Collectively, these results deepen our understanding of psychedelic compounds and offer important directions for development of novel therapeutics.

# Materials and methods

## Participants

Participants were recruited through advertisements placed in local universities from March to July 2015 and underwent a screening visit before inclusion in the larger study protocol (*Preller et al., 2017*). All included subjects were healthy according to medical history, physical examination, blood analysis, and electrocardiography. The Mini-International Neuropsychiatric Interview (MINI-SCID) (*Sheehan et al., 1998*), the DSM-IV self-rating questionnaire for Axis-II personality disorders (SCID-II) (*Fydrich et al., 1997*), and the Hopkins Symptom Checklist (SCL-90-R ) (*Franke, 1995*) were used to exclude subjects with present or previous psychiatric disorders or a history of major psychiatric disorders in first-degree relatives. Participants were asked to abstain from the use of any prescription or illicit drugs for a minimum of two weeks prior to the first test day and for the duration of the entire study, and to abstain from drinking alcohol for at least 24 hr prior to test days. Urine tests and self-report questionnaires were used to verify the absence of drug and alcohol use. Urine tests were also used to exclude pregnancy. Participants were furthermore required to abstain from smoking for at least 60 min before MRI assessment and from drinking caffeine during the test day. Further exclusion criteria included left-handedness, poor knowledge of the German language, cardiovascular disease, history of head injury or neurological disorder, history of alcohol or illicit drug dependence, magnetic resonance imaging (MRI) exclusion criteria including claustrophobia, and previous significant adverse reactions to a hallucinogenic drug.

Twenty-five participants took part in the study. One subject was excluded due to failure in registration caused by an improper head position. Therefore a sample of 24 participants was included in the final analysis (n = 19 males and n = 5 females; mean age = 25.00 years; standard deviation (SD) = 3.60 years; range 20 – 34 years). All participants provided written informed consent statements in accordance with the declaration of Helsinki before participation in the study. Subjects received written and oral descriptions of the study procedures, as well as details regarding the effects and possible risks of LSD and Ket treatment. The Swiss Federal Office of Public Health, Bern, Switzerland, authorized the use of LSD in humans, and the study was approved by the Cantonal Ethics Committee of Zurich (KEK-ZH_No: 2014_0496). The study was registered at ClinicalTrials.gov (NCT02451072). No substantial side effects were recorded during the study. Four participants reported transient headaches after drug effects had worn off. One participant reported transient

sleep disturbances for the first two nights after drug administration. Participants were contacted again three months after the last drug administration. No further side effects were recorded.

## Study design

The study employed a fully double-blind, randomized, cross-over design (see *Figure 1—figure supplement 2*). Randomization was completed by a study nurse who had no other role in the trial. Sample size (n = 24) was determined based on a previous study reporting LSD-induced effects on functional brain connectivity (*Tagliazucchi et al., 2016*). Recruitment was stopped after the determined sample size was reached. Specifically, participants received either:

(i) placebo +placebo (Pla) condition: placebo (179 mg Mannitol and Aerosil 1 mg po) after pretreatment with placebo (179 mg Mannitol and Aerosil 1 mg po);

(ii)placebo +LSD (LSD) condition: LSD (100 µg po) after pretreatment with placebo (179 mg Mannitol and Aerosil 1 mg po), or

(iii) Ketanserin +LSD (Ket+LSD) condition: LSD (100 µg po) after pretreatment with the 5-HT$_{2A}$ antagonist Ket (40 mg po) at three different occasions two weeks apart.

Pretreatment with placebo or Ket occurred 60 min before treatment with placebo or LSD. The resting-state scan was conducted 75 and 300 min after treatment administration. Participants were asked to not engage in repetitive thoughts such as counting and close their eyes during the resting state scan. Compliance to this instruction was monitored online using eye tracking (NordicNeuroLab VisualSystem, http://www.nordicneurolab.com/). The 5D-ASC (a retrospective self-report questionnaire) (*Dittrich, 1998*) was administered to participants 720 min after drug treatment to assess subjective experience after drug intake. In addition, a short version of the 5D-ASC was administered 180, 250, and 360 min after drug treatment to assess the time course of subjective experience.

## Neuroimaging data acquisition

Magnetic resonance imaging (MRI) data were acquired on a Philips Achieva 3.0T whole-body scanner (Best, The Netherlands). A 32-channel receive head coil and MultiTransmit parallel radio frequency transmission was used. Images were acquired using a whole-brain gradient-echo planar imaging (EPI) sequence (repetition time = 2,500 ms; echo time = 27 ms; slice thickness = 3 mm; 45 axial slices; no slice gap; field of view = 240 × 240 mm$^2$; in-plane resolution = 3 × 3 mm; sensitivity-encoding reduction factor = 2.0). 240 volumes were acquired per resting state scan resulting in a scan duration of 10 mins. Additionally, two high-resolution anatomical images were acquired using T1-weighted and T2-weighted sequences. T1-weigthed images were collected via a 3D magnetization-prepared rapid gradient-echo sequence (MP-RAGE) with the following parameters: voxel size = 0.7×0.7×0.7 mm$^3$, time between two inversion pulses = 3123 ms, inversion time = 1055 ms, inter-echo delay = 12 ms, flip angle = 8°, matrix = 320×335, field of view = 224×235 mm$^2$, 236 sagittal slices. Furthermore T2-weighted images were collected using via a turbo spin-echo sequence with the following parameters: voxel size = 0.7×0.7×0.7 mm$^3$, repetition time = 2500 ms, echo time = 415 ms, flip angle = 90°, matrix = 320×335, field of view = 224×235 mm$^2$, 236 sagittal slices.

## Preprocessing

Structural and functional MRI data were first preprocessed according the methods provided by the Human Connectome Project (HCP, RRID:SCR_006942), outlined below, and described in detail by the WU-Minn HCP consortium (*WU-Minn HCP Consortium et al., 2013*). These open-source HCP algorithms, optimized for our specific acquisition parameters and Yale's High Performance Computing resources, represent the current state-of-the-art in distortion correction, registration, and maximization of high-resolution signal-to-noise (SNR). Here we briefly describe the processing steps. Complete details are outlined by Glasser and colleagues (*WU-Minn HCP Consortium et al., 2013*).

First, the T1w/T2w images were corrected for bias-field distortions and warped to the standard Montreal Neurological Institute-152 (MNI-152) brain template through a combination of linear and non-linear transformations using the FMRIB Software Library (FSL, RRID:SCR_002823) linear image registration tool (FLIRT) and non-linear image registration tool (FNIRT) (*Jenkinson et al., 2002*). Then, FreeSurfer's recon-all pipeline was employed to compute brain-extraction, within-subjects registration, and individual cortical and subcortical anatomical segmentation (*Reuter et al., 2012*). T1w/

T2w images were then converted to the Connectivity Informatics Technology Initiative (CIFTI) volume/surface 'grayordinate' space.

Raw BOLD images were first corrected for field inhomogeneity distortion, phase encoding direction distortions and susceptibility arts using the pair of reverse phase-encoded spin-echo field-map images implemented via FSL's TOPUP algorithm (*Andersson et al., 2003*). Motion-correction was then performed by registering each volume in a run to the corresponding single-band reference image collected at the start of each run. BOLD images were then registered to the structural images via FLIRT/FNIRT, and a brain-mask was applied to exclude signal from non-brain tissue. After processing in NIFTI volume space, BOLD data were converted to the CIFTI gray matter matrix by sampling from the anatomically-defined gray matter ribbon.

Following these minimal HCP preprocessing steps, a high-pass filter (>0.008 Hz) was applied to the BOLD time series in order to remove low frequencies and scanner drift. In-house MATLAB (RRID: SCR_001622) tools were used to compute the average variation in BOLD signal in the ventricles and deep white matter. This signal was regressed out of the gray matter time series as a nuisance variable because any BOLD signal change in those structures was likely due to pervasive rather than cortical activity. Finally, mean gray matter time series (i.e. the global signal) was also regressed to address spatially pervasive artefacts, such as respiration. There still remains considerable controversy regarding the utility of mean signal de-noising strategies (*Power et al., 2017*; *Yang et al., 2016b*), with clear pros/cons. While there are several emerging approaches in the literature that attenuate and/or remove sources of global artefacts in BOLD data (*Glasser et al., 2018*), the field-wide gold-standard approach still uses a univariate framework for removing variance from each grayordinate's time series by computing the mean across grayordinates and regressing it from each grayordinate's time course (*Power et al., 2018*). GSR was performed using these standard procedures, explicitly excluding ventricles and white matter (which are defined as separate nuisance regressors). The GS and its first derivative (with respect to time) were used as nuisance predictor terms within a multiple linear regression model along with other nuisance predictor terms (ventricular signal, white matter signal, movement parameters, and the first derivatives of each of these, as noted above). Finally, all data were motion-scrubbed as recommended by *Power et al. (2013)*. As accomplished previously (*Anticevic et al., 2012*), all image frames with possible movement-induced artual fluctuations in intensity were identified via two criteria: first, frames in which the sum of the displacement across all six rigid body movement correction parameters exceeded 0.5 mm (assuming 50 mm cortical sphere radius) were identified. Second, root mean square (RMS) of differences in intensity between the current and preceding frame was computed across all voxels and divided by mean intensity. Frames in which normalized RMS exceeded 1.6 times the median across scans were identified. The frames flagged by either criterion, as well as the one frame preceding and two frames following each flagged frame, were marked for exclusion (logical or). Subjects with more than 50% frames flagged were completely excluded from all analyses. All the included subjects in the final samples passed these criteria.

## Global brain connectivity calculation

Most connectivity studies focus on pre-defined areas (i.e. seed-based approaches). Such approaches assume 'dysconnectivity' across similar regions or networks. However, functional dysconnectivity induced by LSD, especially across heterogeneous associative cortical circuits, may exhibit variability across people. To address this, here we applied recently optimized neuroimaging analytic techniques to identify dysconnectivity in a data-driven fashion, termed global brain connectivity (GBC) (*Anticevic et al., 2013*; *Anticevic et al., 2014b*; *Cole et al., 2011*). GBC is a measure that examines connectivity from a given grayordinate(or area) to all other voxelgrayordinates (or areas) simultaneously by computing average connectivity strength – thereby producing an unbiased approach as to the location of dysconnectivity. Also, unlike typical seed approaches, GBC involves one statistical test per grayordinates (or area) rather than one test per grayordinate-to-grayordinate pairing, substantially reducing multiple comparisons. These improvements dramatically increase the chances of identifying pharmacologically-induced dysconnectivity, or individual differences correlated with symptoms, as we demonstrated by our prior studies conducted in clinical populations (*Cole et al., 2010*; *Anticevic et al., 2013*; *Cole et al., 2011*). By extension, this approach can be readily applied to pharmacological neuroimaging studies. Specifically, the GBC approach (*Cole et al., 2010*; *Cole et al., 2011*) was applied using in-house Matlab tools

studies (*Anticevic et al., 2013*; *Anticevic et al., 2014b*; *Cole et al., 2011*), extended across all grayordinates in the brain, as defined via the CIFTI image space, which was obtained via an adapted version of FreeSurfer software fine-tuned by the HCP pipelines (*Fischl et al., 2002*). Finally, for each grayordinate in the CIFTI image space, we computed a correlation with every other whole-brain grayordinate, transformed the correlations to Fisher z-values, and finally computed their mean. This calculation yielded a GBC map for each subject where each grayordinates value represents the mean connectivity of that grayordinate with all other grayordinates in the brain. We also verified that differences in variance of BOLD signals did not drive our GBC results, as predicted by our prior computational modeling work (*Yang et al., 2014*). To this end, we computed GBC using a non-normalized covariance measure, which did not alter effects. Appropriate whole-brain type I error correction was computed via FSL's PALM tool (see second - Level Group Comparisons below).

## Thalamic seed functional connectivity

To examine the thalamus coupling with all grayordinates in the brain in session one we computed a seed-based thalamus correlation and covariation map by extracting average time-series across all grayordinates in each subject's bilateral thalamus and then correlating/covarying these with each grayordinate. For details on this approach see (*Anticevic et al., 2014a*).

## Global signal regression

Because of emerging findings suggesting that clinical populations exhibit elevated GS variability (*Yang et al., 2014*), we separately examined results without GSR implemented. This demonstration is particularly important given recent reports suggesting that the GS may be abnormally altered in specific clinical populations (*Yang et al., 2014*; *Gotts et al., 2013*), but also that it may contain major elements of respiratory artefacts (*Power et al., 2017*), which could influence GBC analyses.

## Global gray matter signal beta map calculation

To obtain global signal (GS) beta values, we first performed GS regression (GSR) using standard widely adopted procedures (*Anticevic et al., 2013*; *Cole et al., 2011*). The GS timeseries for each subject was obtained by calculating mean raw BOLD signal averaged over all grayordinates for each time point, explicitly excluding ventricles and white matter signal. This GS timeseries was used as nuisance predictor term within a multiple linear regression model. More formally, we used the following multiple regression analysis:

$$BOLD_k^{raw}(t) = b_0 + \sum_{i=1}^{n} b_i X_i + BOLD_k^{preprocessed}(t),$$

where $BOLD_k^{raw}(t)$ represents the raw BOLD signal in grayordinate $k$ as a function of time, $t$. $b_0$ is the intercept, $X_i$ represents the $i^{th}$ nuisance (e.g. GS at that time point), $b_i$ is the corresponding beta weight computed for regressor $X_i$. The last term is the residual signal that is not accounted for by the regressors. In other words, the residual represents the preprocessed BOLD signal at grayordinate $k$. In our model the regressor of interest is GS(t).

$$BOLD_k^{raw}(t) = b_0 + b_{GS}GS(t) + BOLD_k^{preprocessed}(t),$$

The GS beta weights reported are represented by the $b_{GS}$ values obtained from this multiple regression. GS(t) is the spatial average of time-varying BOLD signal across all gray matter grayordinates:

$$GS(t) = \frac{\sum_k^m BOLD_k(t)}{m}$$

The 'mean GS beta weight' computation in *Figure 3* is done by fitting a generalized linear model (GLM) to each grayordinate's BOLD time series to obtain the GS beta weight ($b_{GS}$) as shown above. In that sense, the grayordinate-wise whole-brain map of GS beta weights is more interpretable as a task-evoked GLM analysis than to a functional connectivity measure. In other words, GS beta weights are *not* functional connectivity values and should not be interpreted as such – instead they represent

the amount of GS variance accounted for by that grayordinate for a given subject. This 'GS beta map' was then entered into a second level analysis as done for the functional connectivity dependent measures. This comparison tests the hypothesis that the spatial contribution to the GS is altered under LSD vs. Pla, as done in our prior work (*Yang et al., 2016b*).

## Quality assurance analyses

For quality assurance purposes we computed the following measures: (i) signal-to-noise ratio (defined as mean signal over the entire BOLD time series for a given grayordinate divided by its standard deviation), and (ii) the percentage of 'scrubbed' images. In turn, we correlated these measures with mean Fz-connectivity with and without GSR for the first and second session in the LSD condition (*Figure 1—figure supplement 3*). All correlations were non-significant indicating that changes in GBC induced by LSD are not attributable to motion and image arts.

## Second level statistical analysis

GBC maps for each subject, condition, and session were entered into a 2 × 3 repeated-measures ANOVA and tested using FSL's Permutation Analysis of Linear Models (PALM,[*Winkler et al., 2014*]). Threshold-free cluster enhancement (TFCE) was used to avoid the need to define clusters using arbitrary thresholds for cluster size. We report the default TFCE parameters that were used in the permutation, which are fully described in the PALM code (https://github.com/andersonwinkler/PALM/blob/master/palm_defaults.m). The statistical images were thresholded at p<0.05 (family-wise error corrected), with 10000 permutations. For further analysis connectivity strength (Fz) values for hyper- and hypo-connected areas (based on the LSD vs. (Ket+LSD)+Pla contrast) were averaged across grayordinates for each participant and condition. Furthermore, Fz values for grayordinates within seven functionally-defined networks using parcellations derived by *Yeo et al. (2011)*, *Buckner et al. (2011)* and *Choi et al. (2012)* were averaged for each participant and condition. All analyses were performed with and without GSR. Results were visualized using the Connectome Workbench software (https://www.humanconnectome.org/software/connectome-workbench.html).

## Statistical analysis of behavioral data

The 5D-ASC comprises 94 items that are answered on visual analogue scales (*Dittrich et al., 2006*). Scores were calculated for 11 recently validated scales (*Studerus et al., 2010*): experience of unity, spiritual experience, blissful state, insightfulness, disembodiment, impaired control and cognition, anxiety, complex imagery, elementary imagery, audio-visual synesthesia, and changed meaning of percepts. The short version of the 5D-ASC includes the 45 items that comprise the spiritual experience, blissful state, disembodiment, elementary imagery, and changed meaning of percepts scales. 5D-ASC score was analyzed using a repeated-measures ANOVA with treatment condition (Pla, LSD, and Ket+LSD) and scale as within-subject factors. 5D-ASC short-version score was analyzed using a repeated-measures ANOVA with treatment condition (Pla, LSD, and Ket+LSD), scale, and time (180, 250, and 360 min) as within-subject factors. The 5D-ASC short-version scores of one participant could not be analyzed due to missing data at 360 min after administration. Bonferroni-corrected Pearson correlations were conducted to investigate the relationship between Fz values within the seven functionally defined networks at session two and subjective drug effects (5D-ASC short version at 250 min).

## Gene expression preprocessing

To relate LSD-related neuroimaging effects to the cortical topography of gene expression for candidate receptors, we used the Allen Human Brain Atlas (AHBA, RRID:SCR_007416). The AHBA is a publicly available transcriptional atlas containing gene expression data, measured with DNA microarrays, that were sampled from hundreds of histologically validated neuroanatomical structures across six normal post-mortem human brains (*Hawrylycz et al., 2012*). All reported analyses were performed on group-averaged gene expression maps in the left cortical hemisphere, which were generated following a previously reported procedure (*Burt et al., 2018*). In brief, a group-averaged, dense cortical expression map was constructed through a neurobiologically informed approach using a surface-based Voronoi tessellation combined with a 180-area unilateral parcellation with the

Human Connectome Project's Multi-Modal Parcellation (MMP1.0) (*Glasser et al., 2016*) (*Figure 9— figure supplement 1*).

## Acknowledgements

This research was financially supported by grants from the Swiss National Science Foundation (SNSF, P2ZHP1_161626, KHP), the Swiss Neuromatrix Foundation (2015 – 0103, FXV), the Usona Institute (2015 – 2056, FXV), the NIH(R01MH112746, JDM; DP5OD012109, AA; R01MH108590, AA), the NIAA ( P50AA012870-16, AA & JHK), the NARSAD Independent Investigator Grant (AA), the Yale CTSA grant (UL1TR000142 Pilot Award, AA), and the Slovenian Research Agency (ARRS J7-6829 & ARRS J7-8275, GR).

## Additional information

### Competing interests

Grega Repovs, John D Murray, Alan Anticevic: consults for and holds equity with Blackthorn Therapeutics. The other authors declare that no competing interests exist.

### Funding

| Funder | Grant reference number | Author |
|---|---|---|
| Schweizerischer Nationalfonds zur Förderung der Wissenschaftlichen Forschung | P2ZHP1_161626 | Katrin H Preller |
| Javna Agencija za Raziskovalno Dejavnost RS | ARRS J7-6829, ARRS J7-8275 | Grega Repovs |
| National Institute on Alcohol Abuse and Alcoholism | P50AA012870-16 | Alan Anticevic John H Krystal |
| National Institutes of Health | R01MH112746 | John D Murray |
| Swiss Neuromatrix Foundation | 2015-0103 | Franz X Vollenweider |
| Usona Institute | 2015-2056 | Franz X Vollenweider |
| National Institutes of Health | DP5OD012109 | Alan Anticevic |
| National Institutes of Health | R01MH108590 | Alan Anticevic |
| National Alliance for Research on Schizophrenia and Depression | Independent Investigator Grant | Alan Anticevic |
| Yale CTSA grant | UL1TR000142 Pilot Award | Alan Anticevic |

The funders had no role in study design, data collection and interpretation, or the decision to submit the work for publication.

### Author contributions

Katrin H Preller, Conceptualization, Data curation, Formal analysis, Funding acquisition, Investigation, Writing—original draft, Project administration, Writing—review and editing; Joshua B Burt, Formal analysis, Methodology, Writing—review and editing; Jie Lisa Ji, Methodology, Writing—review and editing; Charles H Schleifer, Brendan D Adkinson, Formal analysis, Writing—review and editing; Philipp Stämpfli, Investigation, Methodology, Writing—review and editing; Erich Seifritz, Resources, Supervision, Writing—review and editing; Grega Repovs, Software, Methodology, Writing—review and editing; John H Krystal, Conceptualization, Resources, Writing—review and editing; John D Murray, Supervision, Visualization, Methodology, Writing—review and editing; Franz X Vollenweider, Conceptualization, Supervision, Funding acquisition, Project administration, Writing—review and editing; Alan Anticevic, Conceptualization, Resources, Data curation, Software, Supervision, Methodology, Writing—original draft, Project administration, Writing—review and editing

Author ORCIDs
Katrin H Preller http://orcid.org/0000-0003-0413-7672
Jie Lisa Ji http://orcid.org/0000-0002-6280-9070
John D Murray https://orcid.org/0000-0003-4115-8181
Alan Anticevic http://orcid.org/0000-0002-4324-0536

### Ethics
Clinical trial registration The study was registered at ClinicalTrials.gov (NCT02451072).
Human subjects: All participants provided written informed consent statements in accordance with the declaration of Helsinki before participation in the study. Subjects received written and oral descriptions of the study procedures, as well as details regarding the effects and possible risks of LSD and ketanserin treatment. The Swiss Federal Office of Public Health, Bern, Switzerland, authorized the use of LSD in humans, and the study was approved by the Cantonal Ethics Committee of Zurich (KEK-ZH_No: 2014_0496).

### Decision letter and Author response
Decision letter https://doi.org/10.7554/eLife.35082.sa1
Author response https://doi.org/10.7554/eLife.35082.sa2

## Additional files
### Supplementary files
- Transparent reporting form
- Reporting standard 1.

### Data availability
All fMRI global brain connectivity maps have been deposited to a Bitbucket repository (https://bitbucket.org/katrinpreller/lsd-effects-on-global-brain-connectivity; copy archived at https://github.com/elifesciences-publications/katrinpreller). Source data files have been provided for Figure 5.

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
