## [Decision Letter]

Thank you for submitting your article "Changes in global brain connectivity in LSD-induced altered states are attributable to the 5-HT2A receptor" for consideration by *eLife*. Your article has been reviewed by three peer reviewers, and the evaluation has been overseen by Laurence Hunt as the Reviewing Editor and Timothy Behrens as the Senior Editor. The reviewers have opted to remain anonymous.

The reviewers have discussed the reviews with one another and the Reviewing Editor has drafted this decision to help you prepare a revised submission.

Summary:

The study examines the effect of 100 micrograms oral LSD administration on resting-state functional MRI data. This has significant effects on Global Brain Connectivity, a measure used to index changes in functional connectivity between regions. The study's key contribution is to pinpoint the pharmacological mechanism of this effect, by showing that it is largely removed by co-administration of 40mg oral administration of the 5-HT2A receptor antagonist ketanserin. Results show that both the subjective and the connectivity changes associated with LSD were blocked fully by ketanserin. Global brain connectivity patterns associated with LSD were also closely linked to gene expression maps of 5-HT2A receptors.

All reviewers were in agreement that this is an unusual and interesting dataset, in terms of the sophisticated 5-HT pharmacology and comparison with gene expression profiles. The reviewers applaud the authors for their use of advanced HCP pipelines. By investigating the 5-HT2A receptor contributions of LSD-induced rsfMRI signal changes, the authors address a relevant topic in psychiatric diseases such as psychosis.

However, all reviewers raised questions about some of the results and interpretations, particularly with respect to the effects of global signal regression on their conclusions.

Essential revisions:

1) The authors show that LSD effects on GBC differ considerably depending on whether global signal regression (GSR) is used during preprocessing. This finding is interpreted to suggest that the results after global signal regression are "correct". The reviewers had concerns about this interpretation. The argument about GSR is not settled, and therefore statements to the effect that GSR "is key for separating signal and noise" are incorrect.

If LSD increases the BOLD signal amplitude in large regions of the brain, this interacts with global signal regression and can drive changes in connectivity. For example, the simulation below shows that whole brain positive-negative effects in GBC that appear very similar to the results presented in Figure 1A can be introduced by the interaction between amplitude changes and global signal regression (note that results without GSR are identical because correlation is scale invariant). It is known that large amplitude changes are found in altered states of consciousness (e.g. during sleep, see Horovitz et al., 2008). It is also likely that the subjects' level of arousal is something that would be expected to change in this LSD study.

% Matlab code for simulation:

T1 = rand(1000,100);

T1 = T1 – repmat(mean(T1),1000,1);% demean

T1_GSR = T1-mean(T1,2)*(pinv(mean(T1,2))*T1);

T2 = [T1(:,1:50).*repmat(2,1000,50) T1(:,51:end)];

T2 = T2 – repmat(mean(T2),1000,1);% demean

T2_GSR = T2-mean(T2,2)*(pinv(mean(T2,2))*T2);

GBC = [mean(corr(T1)); mean(corr(T1_GSR)); mean(corr(T2)); mean(corr(T2_GSR))];figure; set(gcf,'Position',[0 0 1000 400])plot(GBC'); xlabel('voxel'); ylabel('GBC'); title('Simulations to test interaction between amplitude changes and GSR');legend({'Original','Original after GSR','increased amp','increased amp after GSR'})set(gca,'fontsize',16)

A key motivation given for using GSR is that it removes potential respiration artefacts. However, the authors have already removed such artefacts using WM and CSF regression. This lends further evidence to the idea that the global signal in this context may actual represent important global neural activity. All results presented in the manuscript must be interpreted in this context; the effects of any large scale changes in neural activity are not visible with GSR and only focal changes above and beyond these main effects remain.

Given this, and the strong effect of global signal regression on the findings, the reviewers suspect that the results are strongly influenced by underlying changes in amplitude. It is very important to disambiguate these effects, as they would fundamentally change both the interpretation of the LSD effect and the interpretation regarding global signal regression. Therefore, the reviewers argue that a detailed investigation of amplitude changes is absolutely critical for this work.

Lastly, the reviewers find the emphasis on GSR in the manuscript misplaced, and detracting from the core study presented in this work. While they applaud the authors for including results with and without GSR, this work is not well placed to draw generalizable conclusions regarding this preprocessing procedure (particularly given the potential confounds of amplitude changes and hemodynamic coupling changes).

The reviewers' concrete recommendations would therefore be as follows:a) Test directly for amplitude effects (i.e. voxel-wise test for temporal standard deviation, or related measure such as fALFF, between conditions).

b) Continue to present pre and post GSR results, but give them equal weight.

c) Remove *all* interpretations of GSR-related differences, and instead clarify that this dataset is not well-suited for the purpose of drawing conclusions regarding GSR.

2) One further issue is that without GSR, changes in connectivity are left lateralised whereas with GSR they are not. If the global signal only represents artefacts which are by definition global, then why when leaving it in the data are the connectivity changes confined to one hemisphere? Similarly, it is unclear how the data in Figure 5 can be used to support the idea that GSR is desirable. Indeed, it appears that when comparing session 1 vs. session 2 values that the connectivity values are more repeatable without GSR. In the DMN and the limbic network when using GSR, connectivity values are of opposite sign in the Pla and Ket+LSD conditions when comparing across sessions suggesting the GSR is detrimental to the repeatability of the measure in this data.

3) Are there any known effects of LSD on hemodynamic coupling? While it is challenging to control for such effects in this study, this topic warrants further discussion.

4) The comparison against gene expression maps is a strong element of this paper, however it is presented almost as an afterthought in much of the manuscript. One reviewer felt that the authors could emphasize this aspect more strongly in the paper (although another reviewer cautioned that the correlations are fairly low). The authors may wish to emphasise this part of their manuscript more strongly.

5) The effects of the 5-HT2A receptor are investigated (both pharmacologically and in terms of gene expression). However, the authors list a number of other receptors that LSD binds to. Therefore, a more critical discussion (in a limitations section) of the scope of this paper and the potential role of other receptors is warranted.

6) Changes in subjective drug effects over time are presented (Figure 3). Can the authors comment on test-retest reliability of these questionnaire measures please. For example, what was the change over time in the placebo group for subjects that took part in the placebo session as the first study session?

7) The first derivative of the global signal is also included as a regressor, which is quite uncommon. Please include a (mathematical or empirical) justification for this.

8) One reviewer commented that the barplots and distributions in Figure 1B, Figure 4B and others like them should be removed and not relied upon for interpretation, as they present circular reasoning (Kriegeskorte et al., Nat Neuro 2009). Plotting the average connectivity in areas that are defined by looking for differences in connectivity will always lead to significantly different barplots.

---

## [Author Response]

Essential revisions:1) The authors show that LSD effects on GBC differ considerably depending on whether global signal regression (GSR) is used during preprocessing. This finding is interpreted to suggest that the results after global signal regression are "correct". The reviewers had concerns about this interpretation. The argument about GSR is not settled, and therefore statements to the effect that GSR "is key for separating signal and noise" are incorrect.

We agree with the reviewers that this claim was too strong. We revised the manuscript accordingly:

- The statement that “GSR is key for separating signal and noise” has been deleted.

- In accordance with the changes suggested below, we added the following statements to the Discussion section:

“However, this dataset is not well-suited for drawing generalizable conclusions about the effects and suitability of GSR for pharmacological neuroimaging research. Future work that experimentally manipulates variables such as breathing rate and vigilance is needed is needed to fully understand the effects of GSR on pharmacological neuroimaging data and definitively separate pharmacological effects that induce ‘global artefact’ versus those affecting ‘global neural signal’ (Glasser et al., 2018).”

- “artefact removal” has been replaced with “GSR” in the subsection “LSD-induced Alterations in GBC Correlate with HTR2A and HTR7 Cortical Gene Expression”.

Additionally, we implemented further analyses as outlined below.

If LSD increases the BOLD signal amplitude in large regions of the brain, this interacts with global signal regression and can drive changes in connectivity. For example, the simulation below shows that whole brain positive-negative effects in GBC that appear very similar to the results presented in Figure 1A can be introduced by the interaction between amplitude changes and global signal regression (note that results without GSR are identical because correlation is scale invariant). It is known that large amplitude changes are found in altered states of consciousness (e.g. during sleep, see Horovitz et al., 2008). It is also likely that the subjects' level of arousal is something that would be expected to change in this LSD study.% Matlab code for simulation:T1 = rand(1000,100);T1 = T1 – repmat(mean(T1),1000,1);% demeanT1_GSR = T1-mean(T1,2)*(pinv(mean(T1,2))*T1);T2 = [T1(:,1:50).*repmat(2,1000,50) T1(:,51:end)];T2 = T2 – repmat(mean(T2),1000,1);% demeanT2_GSR = T2-mean(T2,2)*(pinv(mean(T2,2))*T2);GBC = [mean(corr(T1)); mean(corr(T1_GSR)); mean(corr(T2)); mean(corr(T2_GSR))];figure; set(gcf,'Position',[0 0 1000 400])plot(GBC'); xlabel('voxel'); ylabel('GBC'); title('Simulations to test interaction between amplitude changes and GSR');legend({'Original','Original after GSR','increased amp','increased amp after GSR'})set(gca,'fontsize',16)A key motivation given for using GSR is that it removes potential respiration artefacts. However, the authors have already removed such artefacts using WM and CSF regression. This lends further evidence to the idea that the global signal in this context may actual represent important global neural activity. All results presented in the manuscript must be interpreted in this context; the effects of any large scale changes in neural activity are not visible with GSR and only focal changes above and beyond these main effects remain.

We respectfully disagree with the conclusion that potential respiration artefacts have been removed by using WM and CSF regression. There has been a great deal of recent work on this precise topic. For instance, Power et al., 2017, explicitly investigated the effects of common de-noising strategies on artefact removal and show that WM and CSF regression alone do not remove respiration artefacts. In fact, their work explicitly demonstrates that the respiratory artefact is a phenomenon that has ‘global-like’ properties.

Furthermore, Power et al., 2018, build on this finding and report “…we isolate a second kind of motion-associated signal, a respiratory signal, that occurs across the entire brain”(see Abstract). They go on to report two methods that they state are capable of quantitatively removing global artefact:

“Go Decomposition (GODEC) is a recently developed multivariate technique that can separate spatially widespread (low-rank) signals from spatially focal (sparse) signals in an fMRI dataset (29).” (p.4)

“A univariate approach to removing global fluctuations in fMRI data is to regress the global signal from all time series (here, we use the mean cortical signal as the global signal, which, as mentioned above, is essentially identical to the whole-brain signal that is often used for such regression purposes).” (p. 5).

We used the later in this paper as the basis for our global artefact removal. We appreciate the reviewers’ point however that this may not have been clear. We now clarify this point in the revised version of the paper:

“While there are several emerging approaches in the literature that attenuate and/or remove sources of global artefact in BOLD data (Glasser et al., 2018), the field-wide gold-standard approach still uses a univariate framework for removing variance from each voxel’s time series by computing the mean across grey matter voxels and regressing it from each voxel’s time course (Power et al., 2018).”

Given this, and the strong effect of global signal regression on the findings, the reviewers suspect that the results are strongly influenced by underlying changes in amplitude. It is very important to disambiguate these effects, as they would fundamentally change both the interpretation of the LSD effect and the interpretation regarding global signal regression. Therefore, the reviewers argue that a detailed investigation of amplitude changes is absolutely critical for this work.

We fully agree with the reviewers that disambiguating the effects of global artefact versus global neural signal, especially in the context of pharmacological neuroimaging, is vital. Given this, we chose to present results both with GSR and without. Furthermore, we agree that the additional suggested analyses may shed father information on how global signal BOLD properties change following LSD administration.

Lastly, the reviewers find the emphasis on GSR in the manuscript misplaced, and detracting from the core study presented in this work. While they applaud the authors for including results with and without GSR, this work is not well placed to draw generalizable conclusions regarding this preprocessing procedure (particularly given the potential confounds of amplitude changes and hemodynamic coupling changes).The reviewers' concrete recommendations would therefore be as follows:a) Test directly for amplitude effects (i.e. voxel-wise test for temporal standard deviation, or related measure such as fALFF, between conditions).

This is an excellent point which we weighed upon initial submission. We agree with the reviewers that this study is not the ideal forum to put forth conclusions about the optimal choices for processing/de-noising methods in pharmacological neuroimaging in general and LSD-induced changes in particular. Nevertheless, we value that the reviewers found merit in presenting this important processing nuance because the results vary dramatically as a function of mean global signal removal (GSR). To help interpret these discrepancies, we have implemented the proposed analysis and expanded the analytic treatment of the GSR steps considerably based on this feedback. Importantly, throughout the revised paper, we present the two methods on equal footing on the same scale and in matched figure format, as requested below. Please note that we updated the color scale to a continuous format to present the effects more intuitively.

Specifically, we conducted three analyses that collectively incorporate the reviewers’ feedback and provide further treatment of this issue:

i) We investigated if the amplitude of the BOLD signal is influenced by GSR across different experimental conditions (i.e. LSD vs. Placebo). Specifically, we quantified ‘amplitudes’ using a measure of local voxel-wise variance – an approach validated in our prior work in the context of clinical neuroimaging and effects of GSR in such datasets (Yang et al., 2014).

ii) We calculated the mean variance of the global signal across all grey matter voxels (as opposed to local voxel-wise variance). We achieved this by defining the mean of all gray matter signal for a given subject based on their FreeSurfer segmentation and then computing the variance of the BOLD signal time course, averaged over all voxels in this global greymatter mask.

iii) To investigate the possibility that the GS itself may exhibit a shifted topography on LSD, as shown in prior work (Yang et al., 2016), we also computed the beta map of the global signal for each subject. In turn, this map allowed us compare the spatial topography of GS under LSD versus placebo conditions and allowed us to test the hypothesis that the GS itself is shifted spatially by LSD. Put differently, it could be possible that on LSD different regions are contributing to the GS map, which may explain the effect of this processing step on the GBC analyses.

Collectively, these three approaches examine the possibility of local versus global BOLD variance changes as well as GS spatial topography shifts under LSD. We present the three analyses below for reviewers’ inspection, which we now added to the revised paper.

i) Testing the hypothesis of voxel-wise variance changes for the LSD > Pla contrast (presented un-thresholded for full inspection).

The Z map presented in Figure 3—figure supplement 1 shows the change in local voxel-wise variance under LSD vs. placebo. The effect illustrates a very weak alteration in local variance (min/max Z = -1.54/+2.28). No effects survived whole-brain correction and no peaks were evident. This result in not consistent with the possibility that LSD markedly alters voxel-wise amplitudes/variance relative to placebo.

ii) Mean grey matter variance across all voxels for the LSD > Pla contrast.

Author response image 1 shows the mean variance across all grey matter voxels under various conditions. Results indicate that variance of the GS does not differ significantly between conditions on average when computing the mean across all grey matter voxels [F(2,46)=0.71, p>0.49)].

iii) Beta map of the GS for the LSD > Pla contrast (presented un-thresholded for full inspection).

Above we show that voxel-wise and average GS variance does not differ markedly for LSD vs. placebo. However, the mean GS analysis above cannot address the possibility that the GS signal itself has a distinct spatial configuration following LSD administration. In other words, which areas are maximally contributing to the mean GS may not be the same after LSD administration. Critically, as noted by the reviewers, these types of analyses are not designed to arbitrate if such a shift is artefactual or neural, but it would provide a key insight regarding the origin of the GBC maps shifts before and after GSR on LSD.

Global Gray Matter Signal Beta Map Calculation.The Z map presented in Figure 3 illustrates a contrast between LSD vs. placebo for the GS beta map that was computed for each subject. Specifically, we calculated the GS (computed as mean grey matter signal) for each frame in the BOLD time course. This mean GS was then used as a regressor in a subject-specific GLM. The resulting beta map indicates which areas are maximally co-varying with the mean GS for each subject under LSD or placebo.

As described in the text, to obtain global signal (GS) beta values, we first performed GS regression (GSR) using standard widely adopted procedures (Cole MWet al. 2011; Anticevic A, MS Brumbaugh, et al. 2012). The GS time series for each subject was obtained by calculating mean raw BOLD signal averaged over all gray matter voxels for each time point, explicitly excluding ventricles and white matter signal. This GS time series was used as nuisance predictor term within a multiple linear regression model. More formally, we used the following multiple regression analysis:BOLDkraw(t)=b0+∑i=1nbiXi+ BOLDkpreprocessed(t),where BOLDkraw(t) represents the raw BOLD signal in voxel *k* as a function of time, *t. b_0_* is the intercept, *X_i_* represents the *i*^th^ nuisance (e.g. GS at that time point), *b_i_* is the corresponding beta weight computed for regressor *X_i_*. The last term is the residual signal that is not accounted for by the regressors. In other words, the residual represents the preprocessed BOLD signal at voxel *k*. In our model the regressor of interest is GS(t).BOLDkrawt=b0+bGSGSt+BOLDkpreprocessedt,

The GS beta weights reported are represented by the *b_GS_* values obtained from this multiple regression. GS(t) is the spatial average of time-varying BOLD signal across all gray matter voxels:GSt=∑kmBOLDk(t)m

The “mean GS beta weight” computation in Figure 3 is done by fitting a generalized linear model (GLM) to each voxel’s BOLD time series to obtain the GS beta weight (bGS). In that sense, the voxel-wise whole-brain map of GS beta weights is more interpretable as a task-evoked GLM analysis than to a functional connectivity measure. In other words, GS beta weights are *not* functional connectivity values and should not be interpreted as such – instead they represent the amount of GS variance accounted for by that voxel for a given subject.

This ‘GS beta map’ was then entered into a 2^nd^ level analysis as done for the functional connectivity dependent measures. This comparison tests the hypothesis that the spatial contribution to the GS is altered under LSD vs. placebo, as done in our prior work (Yang et al., 2016). The result shows LSD>placebo in warm colors and LSD<placebo in cool colors. As evident from Figure 3, the GS beta contrast was quite robust, especially when compared to the local voxel-wise variance results (min/max Z = -5.73/+7.74). Critically, the map revealed a bi-direction spatial shift of the GS under LSD where associative cortices and large areas of sub-cortex showed an elevated GS contribution. In contrast, the blue areas showed a reduced GS contribution under LSD. This map correlated highly with the spatial organization of the LSD-induced changes on GBC. This is unsurprising as GBC is highly sensitive to mean shared signal shifts. Put differently, a GBC matrix will be sensitive to the change in the mean shared signal across the brain. If LSD is altering this mean shared signal topography, then the GBC effect should be similarly affected. To quantify this we calculated the relationship between the LSD-placebo contrast GBC map before and after GSR and the LSD-placebo contrast GS beta map.

This result provides evidence consistent with the hypothesis that LSD induces a transformation in the GS beta map itself, which is contributing the GBC effect pre/post GSR. That said, this analysis still does not resolve or help inform the ‘ground truth’ effect of LSD on baseline functional connectivity.

There is a core limitation to the GBC effect in relation to the GS topography inherent to the way it is computed: Specifically, GBC yields the mean shared statistical dependence from a given voxel to all other voxels. This calculation is therefore affected by the shared variance structure across all voxels (i.e. the map of the GS). If this shared GS variance structure is shifted in one condition versus the next, then the GBC calculation will shift accordingly in a spatially ordered way corresponding to the GS spatial shift. This is precisely what the Figure 3 illustrates. We are very thankful that the reviewers’ comments pushed us down this analytic path, which now fundamentally enriches the way one interprets the GBC results in our and prior LSD work.

To frame this problem more precisely, it cannot be determined definitively from the GBC effect if LSD indeed elevates or reduces mean connectivity in associative vs. somato-sensory cortices. This interpretational challenge, which is present in this and prior work, is highlighted by the presented GS beta map analyses because it is not clear if GS beta map transformations on the GBC effect under LSD are primarily neural or artefactual. In other words, the presented results do not provide a ‘ground truth’ of directional manipulations under LSD irrespective of GS-related shifts.

To inform this problem, a complementary analysis is needed, which yields a map that is interpretationally consistent irrespective of GS-related shifts. To construct such a map, here we focused on thalamocortical circuits to provide a well-established and replicated effect in the clinical literature that is not strongly affected by GSR transformations (Anticevic et al., 2014, Yang et al., 2014, Woodward et al., 2012).

To examine thalamic coupling with all voxels in the brain in session 1 we computed a seed-based thalamus map by extracting average time-series across all voxels in each subject's anatomically defined bilateral thalamus (via FreeSurfer segmentations) and then quantifying its relationship with each gray matter voxel. Critically, we used both the correlation and covariance as methods of statistical dependence, which we conjuncted (Cole et al., 2016). We did this because covariance reflects a non-normalized measure of shared signal co-variation (which is scale free and unaffected by variance structure) whereas correlation is inherently normalized by pooled variance. This yielded a more robust map of thalamic covariation. Furthermore, given that GS induces mean signal shifts (i.e. it may induce anti-correlation), here we also obtained the top and bottom 10% of all thalamic connections from the mean connectivity (correlation and covariance) maps before and after GSR. This final 4-way conjunction map ensured that the resulting regions in the top range exhibit thalamic FC irrespective of processing (i.e. GSR/noGSR) or statistical method (i.e. r or cov). This map was then used to calculate LSD induced effects on top 10% and bottom 10% of the connections. The prediction was that LSD would decrease connections that were in the top 10% (i.e. highly positive thalamic connections at baseline, which represent thalamo-associative FC). In turn, we predicted that LSD would elevate connections that were in the bottom 10% (i.e. highly weak thalamic connections at baseline, which represent thalamo-sensory FC).

Figure 4 illustrates the result of these analyses. Panel A shows the baseline thalamic FC computed after GSR for the correlation method, illustrating the expected patterns (i.e. weak thalamo-sensory FC and strong thalamo-associative FC). Panels B and C show the difference in the average signal between drug conditions for the correlation method after GSR. Here, LSD consistently decreases FC in associative areas and increases FC in somato-sensory regions (panel A) and this effect was preserved for the LSD-(Ket+LSD) analysis (panel B). Without GSR however, inconsistent results are revealed (Figure 4—figure supplement 2).

Therefore, to reconcile the interpretation of the LSD effect directionality we performed a conjunction approach that was robust to processing method and statistical approach. As noted, the top and bottom 10% of all connections were extracted from the mean connectivity (correlation and covariance) maps in the Pla condition (before and after GSR) were used to compute a conjunction map providing the strongest and weakest thalamic connections irrespective of analysis method (panels D and E). This conjunction map was then used as a mask to extract the average signal across these regions in the LSD-Pla and LSD-(Ket+LSD) contrast before and after GSR in the seed-based correlation/covariace analyses as well as GBC correlation/covariance analyses. Panel F shows the difference in the average signal between these drug conditions for the different analyses methods (seed FC/GBC, correlation/covariance) after GSR. Here, LSD consistently decreases FC in associative areas and increases FC in somato-sensory regions irrespective of analysis method. Importantly, the thalamic seed analyses reveals a matched GBC effects. Without GSR however (Panel G), seed FC and GBC results are inconsistent. Furthermore, in contrast to results after GSR, LSD did not consistently decrease connections that were in the top 10% or elevate connections that were in the bottom 10% without GSR. To investigate individual differences, we computed the correlation between the top and bottom connections before and after GSR across participants (Panel H, full connectivity matrix is presented in Figure 5—figure supplement 1). The prediction was that individuals with biggest elevation for bottom 10% would should the biggest drop in the top 10% under LSD. Predicted negative individual differences emerged after GSR but without GSR results were not compatible with individual difference predictions based either for thalamic seed analysis or GBC. Collectively, these results are consistent with the hypothesis that, following GS cleanup, LSD reduces shared signals for association cortices but elevates shared signals for sensory and somatomotor areas across both seed-based thalamic and GBC analyses.

b) Continue to present pre and post GSR results, but give them equal weight.

Weagree fully. As noted, we are now giving both analyses equal weight and have introduced additional expansion of the result, as articulated above. In particular, we have now added Figure S5 (displaying the results without GSR) to the main manuscript as Figure 2.

c) Remove all interpretations of GSR-related differences, and instead clarify that this dataset is not well-suited for the purpose of drawing conclusions regarding GSR.

We also agree with this suggestion and have edited the manuscript accordingly. Detailed changes are outlined in response 1.

2) One further issue is that without GSR, changes in connectivity are left lateralised whereas with GSR they are not. If the global signal only represents artefacts which are by definition global, then why when leaving it in the data are the connectivity changes confined to one hemisphere? Similarly, it is unclear how the data in Figure 5 can be used to support the idea that GSR is desirable. Indeed, it appears that when comparing session 1 vs. session 2 values that the connectivity values are more repeatable without GSR. In the DMN and the limbic network when using GSR, connectivity values are of opposite sign in the placebo and ket+LSD conditions when comparing across sessions suggesting the GSR is detrimental to the repeatability of the measure in this data.

These are all thoughtful and relevant points, many of which are now addressed by the added analyses articulated above. That said, we respectfully disagree with the conclusion that changes in connectivity without GSR are exclusively lateralized. The key intuition here emerges when inspecting the un-thresholded maps in Figure S5 (lower panels of Figures S5C-E, now Figure 2). These maps are highly consistent with the hypothesis of a symmetrical drug-induced effect on GBC with and without GSR. However, we agree that the TFCE Type I error protected maps do give the impression that effects are stronger in one hemisphere than the other. This is occurring due to peaks of the effect being marginally higher in one hemisphere relative to the other. To quantify this point, we correlated the L vs. R CIFTI map elements (i.e. L vs. R surfaces) prior to thresholding. This indicates that the effect is indeed highly symmetric (r=0.59 – 0.80).

The second component of the reviewers’ feedback concerns the ability to arbitrate GSR/noGSR effects as more valid. We agree with this feedback. In fact, Figure 5 (now Figure 7) has not been included to support one analysis method over the other. We apologize if this was the impression in the original version of the paper. Actually, Figure 5 is presenting all data while giving equal weight to both methods, as advised by the reviewers.

In addition, considered carefully examined T1 vs. T2 effects, which we agree differ across conditions. We also see how the reviewers might arrive at the conclusion that some results might seem more ‘replicable’ without GSR. We apologize for the lack of clarity in our original version, which perhaps did not fully unpack this intuition. We appreciate the chance to more carefully discuss this effect. This statistical phenomenon of ‘artefactual’ replication is not surprising, when one considers GSR is intended to remove sources of spatially pervasive structured artefact that also likely persists across sessions (e.g. elevated respiratory artefacts). Here there is an important nuance between ‘noise’ and ‘artefact’. Pure ‘white’ noise can be signal-averaged out and would not yield a consistent effect. In contrast, structured artefact represents a signal that can induce the same spurious effect multiple times (Power et al., 2018, Glasser et al., 2018). Put differently, if a structured artefact is large in both measurements (T1 and T2), then this artefact will spuriously drive the effect and therefore will be replicable. We are not claiming that the current data are able to fully disambiguate between these scenarios (i.e. true replication vs. spurious replication). For this reason, we have not expanded our detailed description of this effect, which was presented originally in Figure 4. In turn, we provide a more detailed discussion of the potential underlying pharmacological mechanisms which may relate to this phenomenon, building on the original section “Time-dependent effects of LSD” in the Discussion section. Collectively, we are thankful for this careful feedback which we feel improved arguments surrounding T1/T2 effects, which we now include in the revised version.

3) Are there any known effects of LSD on hemodynamic coupling? While it is challenging to control for such effects in this study, this topic warrants further discussion.

We appreciate the reviewers raising this excellent point. To the best of our knowledge, there is no study investigating the effects of LSD on hemodynamic coupling specifically. However, we agree with the reviewers that this point needs further discussion. Therefore, we added the following paragraph to the Discussion section:

“Furthermore, so far, it has not been investigated whether LSD modulates neurovascular coupling and the hemodynamic response function. […] Furthermore, animal studies which offer the possibility to combine neuronal recordings with simultaneous measurement of hemodynamics are desirable to support the interpretation of functional imaging studies of LSD effects in the human brain.”

4) The comparison against gene expression maps is a strong element of this paper, however it is presented almost as an afterthought in much of the manuscript. One reviewer felt that the authors could emphasize this aspect more strongly in the paper (although another reviewer cautioned that the correlations are fairly low). The authors may wish to emphasise this part of their manuscript more strongly.

We very much appreciate the reviewers’ positive feedback and the recognition of the added value that this analysis brings to the study. We appreciate the comment to perhaps prioritize this effect. The comparison against gene expression maps is presented as the last figure because this is a highly novel analysis that we felt needs further replication before it is featured centrally. Nevertheless, we have now emphasized this effect further, as recommended by the reviewers:

- We highlighted this effect more prominently in the Abstract.- We further emphasized the rationale and the effect in the Introduction section.

- We expanded the Discussion section on gene expression, integrating the caution proposed by the reviewers concerning magnitude of correlations.

5) The effects of the 5-HT2A receptor are investigated (both pharmacologically and in terms of gene expression). However, the authors list a number of other receptors that LSD binds to. Therefore, a more critical discussion (in a limitations section) of the scope of this paper and the potential role of other receptors is warranted.

We agree with the reviewer that from the current study no conclusions can be drawn about the functional contribution of other receptors to the effects of LSD beyond the 5-HT2A receptor, which is afforded by the ketanserin manipulation. We discuss this limitation in the subsection “Session Impacts Global Brain Connectivity in Ketanserin+LSD Condition” and have now added the following paragraph to the Discussion:

“While the current results strongly implicate the involvement of the 5-HT2A receptor in LSD-induced effects, it must be noted that no further conclusions can be drawn regarding the functional contribution of other receptors agonized or antagonized by LSD. This limitation needs further investigation in future studies by blocking serotonin and dopamine receptors involved in the pharmacology of LSD beyond the 5-HT2A receptor. Furthermore, the contribution of these receptors to the effects of different doses of LSD still need to be studied.”

6) Changes in subjective drug effects over time are presented (Figure 3). Can the authors comment on test-retest reliability of these questionnaire measures please. For example, what was the change over time in the placebo group for subjects that took part in the placebo session as the first study session?

This is a thoughtful point and we appreciate the opportunity to add this information. There are two parts to this comment:

i) Test-retest reliability of the subject drug-induced effects over time.

ii) Test-rests reliability of the scale itself in the placebo condition over time.

Regarding the first point, test-retest reliability of these measures is high. Within each drug condition, mean scores over time correlated highly and significantly (all Pearson’s r > 0.40, max r = 0.99). We now quantified this in the LSD condition across the 5 subjective subscales, which we present in the supplement of the revised paper (see Figure 5—figure supplement 1).

Regarding the second point, 8 (out of 24) participants received placebo in the first study session. Within this group the subjective effects were modest at best. For instance, the largest mean change score was observed for the scale “disembodiment” between T1 and T2 and this was a mean=3.5 (SD=9.37) on scale ranging from 0-100. Put simply, the range of observed values in the placebo condition makes is statistically challenging (if not invalid) to compute a test-rests. We are certain the reviewers appreciate the intuition that the scale in question is designed to assay the subjective effects *after* drug administration. Thus, while we see the logic of the placebo group analysis suggestion, the practicality is that this analysis yields little useful variance. Therefore, we now added the test-retest effects for the LSD drug-induced subjective change. We are thankful for the reviewers engaging this issue.

7) The first derivative of the global signal is also included as a regressor, which is quite uncommon. Please include a (mathematical or empirical) justification for this.

We included the first derivative of the global signal to be able to remove potential temporal shifts in breathing artefacts. This procedure is actually common and has been applied in the literature papers (e.g., Fox et al., 2009, Journal of Neurophysiology).

8) One reviewer commented that the barplots and distributions in Figure 1B, Figure 4B and others like them should be removed and not relied upon for interpretation, as they present circular reasoning (Kriegeskorte et al., Nat Neuro 2009). Plotting the average connectivity in areas that are defined by looking for differences in connectivity will always lead to significantly different barplots.

We agree with the reviewers that plotting the average connectivity in areas that are defined by significant differences between conditions will lead to significantly different barplots. Therefore we removed the indication of significance from the barplots (asterisk symbol and legend) in Figure 1, 4 (now Figure 5), and S5 (now Figure 2). However, we respectfully disagree with the request to remove the plot itself, since we argue for presenting the data as transparently and comprehensively as possible. The distribution – and barplots enable the reader to gain additional insight in the nature of alterations in connectivity induced by drugs and/or session beyond brain maps. Furthermore, since Figure 1 and S5 (now Figure 2) represent the main effect of three conditions, these barplots allow the reader to directly compare these three conditions, which is not possible from the main effect brain map.